# The phage L capsid decoration protein has a novel OB-fold and an unusual capsid binding strategy

Rebecca L Newcomer[1], Jason R Schrad[2], Eddie B Gilcrease[3], Sherwood R Casjens[3], Michael Feig[2], Carolyn M Teschke[1]*, Andrei T Alexandrescu[1], Kristin N Parent[2]*

[1]Department of Molecular and Cell Biology, University of Connecticut, Storrs, United States; [2]Department of Biochemistry and Molecular Biology, Michigan State University, East Lansing, United States; [3]Division of Microbiology and Immunology, Department of Pathology, University of Utah School of Medicine, Salt Lake City, United States

**Abstract** The major coat proteins of dsDNA tailed phages (order *Caudovirales*) and herpesviruses form capsids by a mechanism that includes active packaging of the dsDNA genome into a precursor procapsid, followed by expansion and stabilization of the capsid. These viruses have evolved diverse strategies to fortify their capsids, such as non-covalent binding of auxiliary 'decoration' (Dec) proteins. The Dec protein from the P22-like phage L has a highly unusual binding strategy that distinguishes between nearly identical three-fold and quasi-three-fold sites of the icosahedral capsid. Cryo-electron microscopy and three-dimensional image reconstruction were employed to determine the structure of native phage L particles. NMR was used to determine the structure/dynamics of Dec in solution. The NMR structure and the cryo-EM density envelope were combined to build a model of the capsid-bound Dec trimer. Key regions that modulate the binding interface were verified by site-directed mutagenesis.
DOI: https://doi.org/10.7554/eLife.45345.001

*For correspondence:
carolyn.teschke@uconn.edu
(CMT);
kparent@msu.edu (KNP)

**Competing interests:** The authors declare that no competing interests exist.

## Introduction

Viral icosahedral capsids are formed from multiple copies of a single or a few types of highly structurally-conserved coat proteins that encapsidate the genome (*Bamford et al., 2005*). The minimum number of subunits needed to build an icosahedral capsid is 60 coat proteins, and the result is a T = 1 icosahedral geometry. There is a direct correlation between genome size and T-number for dsDNA containing phages. Building a bigger capsid necessitates the use of more coat protein subunits, and requires that chemically identical proteins assemble into capsid sites with different 'quasi-equivalent' conformations (*Caspar and Klug, 1962*). For example, in many of the well-studied double stranded DNA (dsDNA) containing phages of the order *Caudovirales*, the capsids use a T = 7 geometry (see *Figure 1A*). These use 11 vertices formed by coat protein pentons and an additional 60 hexons to create the capsid. The 12th vertex breaks the icosahedral symmetry and is occupied by a portal complex to which a tail that specifies host binding is attached. Estimates predict that $10^{31}$ viruses are in Earth's biosphere (*Bergh et al., 1989*; *Wommack and Colwell, 2000*), with dsDNA containing bacteriophages being the most abundant. These phages have an immense diversity in terms of size and complexity. The ubiquitous HK97-like fold is the building block for virtually all dsDNA containing phages and allows for enormous versatility in icosahedral geometry, that can lead to differences in biophysical properties (*Hua et al., 2017*). To withstand environmental stresses and the internal pressure that amasses as a result of dsDNA genome packaging, some dsDNA phages

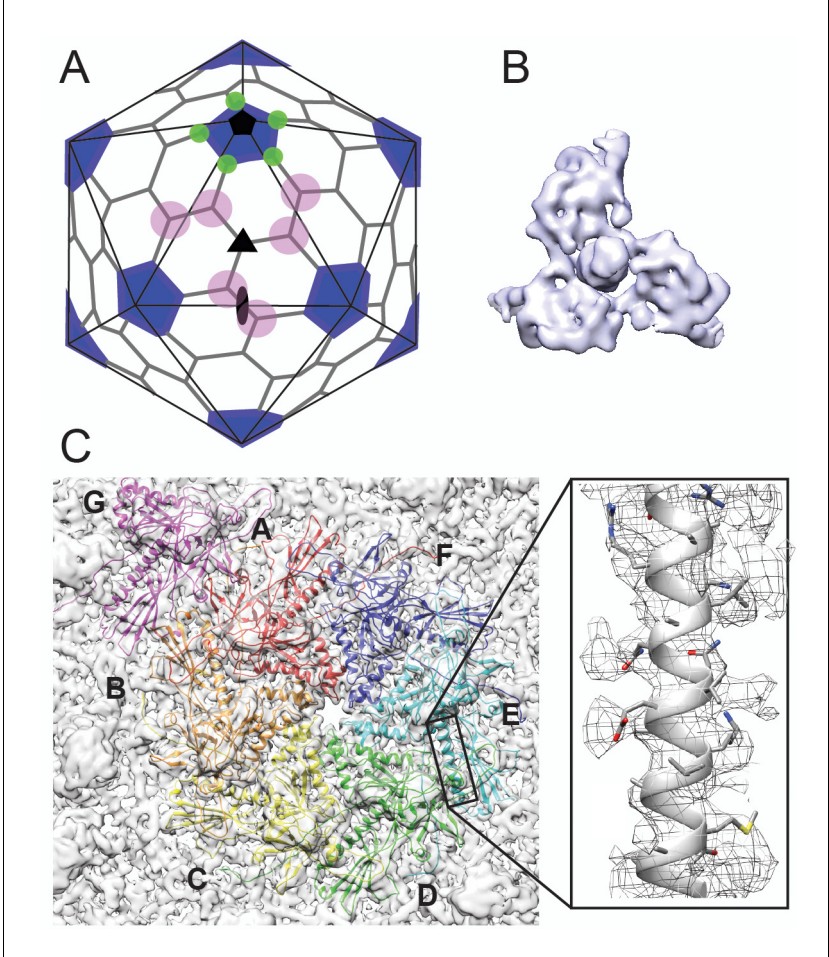

**Figure 1.** Cryo-EM imaging and icosahedral image reconstruction of the mature phage L capsid. (**A**) Schematic of a T = 7 icosahedral capsid overlaid with a black icosahedral cage. The different symmetry axes are marked; two-fold by a black oval, five-fold by a black pentagon, and icosahedral three-fold by a black triangle. Magenta circles highlight quasi three-fold binding sites between hexamers on one icosahedral facet. The magenta quasi three-folds are the preferred binding sites for Dec. Green circles highlight additional quasi three-fold sites between hexons and pentons surrounding one vertex, to which Dec has never been observed to bind. (**B**) Segmented electron density for a capsid-bound Dec trimer shown in a top-down view. (**C**) Enlarged view of an asymmetric unit showing coat subunits color coded according to chains A-G, where the coat density is segmented away from the Dec trimer. Additionally, an enlarged area is shown highlighting a region of the spine helix to show the goodness of fit of the coat protein homology model.

DOI: https://doi.org/10.7554/eLife.45345.002

The following figure supplements are available for figure 1:

**Figure supplement 1.** Phage L cryo-EM and 3D reconstruction data.
DOI: https://doi.org/10.7554/eLife.45345.003
**Figure supplement 2.** Phage L coat protein subunit.
DOI: https://doi.org/10.7554/eLife.45345.004

encode additional 'decoration' proteins that bind to the exterior of their capsids and stabilize the virions. How various decoration proteins recognize and bind to specific sites on capsids with different icosahedral geometries is, however, still poorly understood.

The tight packing of the dsDNA genomes into the virions of many phages and herpesviruses creates enormous internal pressure (10–60 atm) within the capsids, resulting in high-energy states that prime the particles for infection, and facilitate delivery of the majority of the viral genomes into the hosts (*Evilevitch, 2018*; *Kellermayer et al., 2018*; *Bauer et al., 2013*; *Bauer et al., 2015*; *Sharma et al., 2017*; *Gelbart and Knobler, 2009*; *Grayson et al., 2007*; *Kindt et al., 2001*). Phages

have evolved diverse strategies to strengthen their capsids against the internal pressures resulting from genome packaging. In the well-studied phage HK97, the coat proteins form covalent crosslinks (*Popa et al., 1991*) with neighboring subunits, thereby creating a unique 'chain mail' lattice (*Wikoff et al., 1998*; *Duda, 1998*). By contrast in some other phages, a separately expressed auxilliary 'decoration' protein (sometimes called 'cementing' protein) functions as a molecular staple to stabilize the mature capsids. The essential phage λ decoration protein gpD stabilizes its capsid (*Sternberg and Weisberg, 1977*; *Imber et al., 1980*) and binds at all possible binding sites between capsomers, including positions at both icosahedral and local three-fold symmetry axes (*Dokland and Murialdo, 1993*; *Lander et al., 2008*; *Yang et al., 2008*). Other related phages do not utilize covalent crosslinks for stability, or auxillary decoration proteins. Instead, these capsids rely entirely on stabilizing interactions between protein subunits. For example, a portion of the coat protein called the 'P-loop' can create stabilizing interactions with neighboring capsid subunits around three-fold symmetry axes in the absense of auxiliary decoration proteins for many diverse phages (*Parent et al., 2010*; *Chen et al., 2011*; *Spilman et al., 2011*; *Parent et al., 2012a*; *Parent et al., 2014*).

Bacteriophages P22 and L are members of the short-tailed phages (family *Podoviridae)*, in the sub-group of P22-like phages (*Casjens and Thuman-Commike, 2011*). Phage P22 is a particularly well-studied model for understanding the biophysical mechanisms of capsid assembly (*Parent et al., 2006*) and capsid protein structures at near-atomic level detail (*Teschke and Parent, 2010*; *Parent et al., 2018*). The coat proteins of the P22-like phages, like all known tailed phages and herpesviruses, share a canonical HK97-like coat protein fold, one of the most ubiquitous protein folds in nature (*Hendrix, 2002*; *Baker et al., 2005*). The coat proteins of phage L and phage P22 share 99.6% identity with only four amino acids differing out of 430, but only phage L encodes the small auxillary decoration protein, Dec (*Gilcrease et al., 2005*). Phage L and P22 encapsidate similar length genomes and have similar capsid diameters. Therefore, the internal pressures within both capsids are likely to be similar. Dec binds to P22 capsids in vitro with nanomolar affinity at the same sites as on phage L virions, and increases the stability of P22 capsids at elevated temperatures when treated with a $Mg^{++}$ chelator in vitro (*Gilcrease et al., 2005*; *Tang et al., 2006*). Thus, the role(s) of Dec in the life cycle of phage L compared to P22 is not immediately obvious, and, unlike gpD in phage λ, Dec is not essential to withstand genomic packaging forces in phage L virions (E. Gilcrease and S. Casjens, unpublished). The reasons why some phages require stabilization mechanisms such as decoration proteins, and others do not, despite using highly similar functional capsid building blocks remain poorly understood.

Like phage λ's cementing protein gpD, phage L's Dec binds as a homotrimer only to mature particles, and not precursor procapsids (*Gilcrease et al., 2005*). This indicates that the proteins are not involved in capsid assembly but recognize specific surface topology features of mature capsids. During maturation, dsDNA-containing phage and herpesvirus capsids undergo expansion and massive conformational changes, stabilizing the particles and exposing new residues to the surface (*Chen et al., 2011*; *Teschke and Parent, 2010*; *Veesler and Johnson, 2012*). The T = 7 organization of P22 and phage L capsids results in different protein environments between neighboring capsomers due to the previously described 'quasi-equivalence' constraints (*Figure 1A*). At the center of each of the capsid's 20 icosahedral facets, there is a symmetric trimeric interaction formed by the three surrounding hexons (called the 'icosahedral three-fold' axis; black triangle in *Figure 1A*). Between other hexons there are 'quasi-three-fold' symmetry axes (magenta circles in *Figure 1A*), which have a sublty different arrangement than the icosahedral three-fold symmetry axes. These quasi-three-fold sites are less symmetric and have bent contacts between capsomers when compared with an icosahedral-three-fold symmetry site. Lastly, there is a second type of quasi-three-fold symmetry site that occurs between hexons and the pentons (green circles in *Figure 1A*). Tight binding (~nM affinity) of Dec occurs at only quasi-three-fold axes between hexons (magenta circles). In contrast to other types of cementing proteins, binding of Dec to the 20 'true' icosahedral three-fold symmetry axes (black triangles in *Figure 1A*) occurs only weakly. The binding affinity for the true three-folds is at least an order of magnitude lower than for the quasi-three-folds (*Parent et al., 2012b*), so that the former sites are very sparsely occupied by Dec on the native phage L particles. No binding has been detected at the quasi-three-fold sites surrounding the pentons (green circles) (*Tang et al., 2006*; *Parent et al., 2012b*). Dec's ability to bind discriminately is in sharp contrast with the decoration proteins of other phages that fully saturate all symmetry-related sites in mature virions.

Defining the mechanisms by which different decoration proteins, such as gpD and Dec, bind viral particle surfaces is not only important for understanding the underlying biology, but is also critical for (1) the potential exploitation of phages in nanomedicine (*Tao et al., 2018b*; *Tao et al., 2018a*; *Vernhes et al., 2017*; *Asija and Teschke, 2018*; *Serwer and Wright, 2018*), (2) structure-guided design of virus-inspired nanomaterials (*Sharma et al., 2017*; *Parent et al., 2012b*; *Douglas and Young, 2006*; *Schwarz et al., 2015*; *McCoy et al., 2018*; *Catalano, 2018*), and (3) shedding light on capsid assembly and stabilization processes (*Teschke and Parent, 2010*; *Suhanovsky and Teschke, 2015*). We report the structure of phage L Dec protein, which has a novel fold for a decoration protein and propose an explanation for how Dec may be able to bind to subtly different capsid binding sites. We used a combination of structural approaches to understand how Dec interacts with phage capsids including (1) cryo-electron microscopy (cryo-EM) and three-dimensional image reconstructions (3DR) of native phage L particles, (2) Nuclear Magnetic Resonance (NMR) structure and dynamics of Dec in solution, and (3) computational modeling of the Dec trimer. Dec displays significant asymmetry and flexibility when bound to capsids. The N-terminal globular domain of the protein that contacts the capsid has an <u>O</u>ligonucleotide/oligosaccharide-<u>B</u>inding structure (OB-fold; *Murzin, 1993*), and the C-terminal portion of Dec is comprised of a putative three-stranded β-helix domain. Several key residues within both the phage L coat protein lattice and the Dec trimer are predicted from our structural model to modulate the affinity of Dec binding. The contributions of these residues toward Dec binding were probed by site-directed mutagenesis. Our findings reveal that Dec uses a binding mechanism that is not shared by other known phages and viruses, uncovering new insights into phage biology and stabilization mechanisms. The combination of these results provides a highly tunable and novel platform for future uses in virus-inspired nanodesign.

## Results

### Cryo-EM structure of phage L at near-atomic resolution

As a first step toward understanding the mechanism by which Dec binds to phage L, we determined the structure of native phage L capsids using cryo-EM (*Figure 1* and *Figure 1—figure supplement 1*). Local resolution analysis (*Cardone et al., 2013*) shows that some areas of the map are better resolved than others, including the capsid proteins and the base of the Dec trimer, whereas the distal end of Dec is at much lower resolution (*Figure 1—figure supplement 1C*). Both the overall protein structure, as well as the specific contact points for attachment of Dec trimers to the coat protein subunits are clearly discernable near quasi-three-fold symmetry axes between hexons. As anticipated from previous studies (*Tang et al., 2006*; *Parent et al., 2012b*), Dec very sparsely occupied icosahedral three-fold symmetry sites in phage L virions (data not shown), with only a hint of very weak density observed at these positions and no Dec occupancy at the alternative quasi-three-fold sites between hexons and pentons (*Figure 1A*). Segmentation of the Dec density from the virion density map for subsequent analysis was performed using UCSF Chimera's Segger (*Pintilie et al., 2010*) (*Figure 1B,C*).

### Structures of the phage L coat lattice and capsid-bound Dec trimer

As an initial guide for accurately fitting the phage L coat protein into the cryo-EM density, we used the most recent 3.3 Å-resolution structure of phage P22 coat protein (PDB ID: 5UU5; *Hryc et al., 2017*), since phage L and phage P22 are highly homologous, differing at only four positions of their 430 amino acid coat proteins (*Tang et al., 2006*). Upon initial docking, we found small discrepancies between the phage L capsid cryo-EM density and the P22 coat protein asymmetric unit structure. Therefore, to optimally fit the cryo-EM density we refined the phage L asymmetric unit where each capsid protein was allowed to move independently in the cryo-EM density envelope using the program Phenix (*Table 1*). The phage L capsid protein subunit domains (*Figure 1—figure supplement 2*) are named as defined for P22 coat protein (*Hryc et al., 2017*; *Rizzo et al., 2014*). The four amino acid sequence differences between phage L and P22 coat proteins were accounted for during modeling and are not near the Dec binding interface (see *Video 1*). The R101H difference is within the spine helix pointing towards the capsid interior, I154L is located in the A-domain toward the hexamer center, M267L is in the I-domain, adjacent to but not interacting with Dec, and A276T in also

**Table 1.** Cryo-EM data collection and model refinement statistics.

| Data collection | | |
|---|---|---|
| EM equipment | | FEI Titan Krios |
| Voltage (kV) | | 300 |
| Detector | | DE-20 |
| Pixel size (Å) | | 1.26 |
| Electron Dose (e⁻/Å²) | | 27 |
| Defocus range (μm) | | 0.35–2.49 |
| **Reconstruction** | | |
| Software | | AUTO3DEM |
| Number of particles | | 7879 |
| Symmetry | | 532 |
| Map resolution (Å) at FSC = 0.143 | | 4.2 |
| **Model building and Refinement** | | |
| Software | | COOT, Phenix |
| **Model Statistics and Validation** | **Coat** | **Dec** |
| No. protein chains in ASU | 7 | 3 |
| No. residues per protein | 430 | 134 |
| Model resolution (Å) at FSC = 0.5 | 4.30 | 5.80 |
| Cross-correlation coefficient (CC) | 0.61 | 0.40 |
| MolProbity Score | 2.21 | 1.93 |
| Clashscore | 2.20 | 0.87 |
| **R.m.s. deviations** | | |
| Bond length (Å) | 0.021 | 0.018 |
| Bond angle (°) | 2.62 | 2.21 |
| Preferred | 92.42 | 77.53 |
| Allowed | 5.41 | 13.69 |
| Outlier | 2.17 | 8.59 |

DOI: https://doi.org/10.7554/eLife.45345.005

located in the I-domain but on the distal end pointing toward the center of the hexamer (*Figure 1—figure supplement 2*). Overall, there were only very minor differences between the optimally fit P22 and phage L capsid lattices. The phage L coat protein (Cα backbone) deviated from those in the P22 structure (PDB 5UU5) by about 1 Å for the hexamer and 1.5 Å for the penton unit, which is in the range of what may be expected for structures of homologous proteins (*Eyrich et al., 2001*).

The capsid-bound Dec trimer was previously reported to have a tripod shape with three N-terminal legs interacting with the capsid surface, and a protruding central C-terminal head or stalk based on low-resolution data (*Tang et al., 2006*; *Parent et al., 2012b*). Our results agree with the previously published lower resolution structure; however, the higher resolution data for Dec, particularly in the N-terminal region, now allows a detailed analysis of the fold of the protein and interpretation of the binding interface. The trimeric Dec density (one segmented trimer is shown in *Figure 1B*) is asymmetric and well defined at the quasi-three-fold symmetry axes between hexons (*Parent et al., 2012b*; *Schwarz et al., 2015*). The resolution within the Dec density is on average lower than the capsid, likely due to protein flexibility, especially in the C-terminal stalk region (*Figure 1—figure supplement 1*).

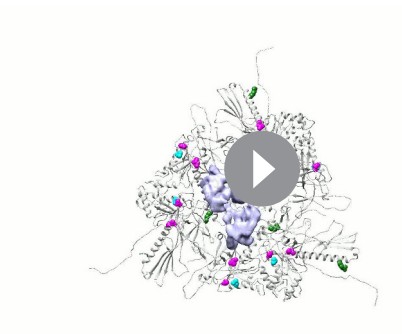

**Video 1.** Location of the amino acid differences between phage L and P22, relative to the Dec binding interface. A single segmented Dec density is shown in lavender, and coat protein subunits that comprise the binding interface are shown as grey ribbons. The amino acid substitutions between phage L and P22 are shown as spheres color-coded to match the protein domains as shown in *Figure 1—figure supplement 2*: R101H (green), I154L (cyan), A276 and M267 (magenta).
DOI: https://doi.org/10.7554/eLife.45345.006

# The NMR structure of the Dec monomer consists of an OB-fold domain and an unfolded C-terminus

To investigate the properties of free Dec in solution, we used NMR to characterize the unassembled protein in the absence of capsids using an acid unfolding protocol of lowering the pH to 2, followed by a refolding step induced by adjusting the pH to four as defined in *Newcomer et al. (2018)*. Subsequent characterization of the protein using size exclusion chromatography, native-gel electrophoresis, and $^{15}$N NMR relaxation measurements (*Figure 3—figure supplement 1*) showed that the unfolding/refolding protocol converts purified Dec from a trimer to partially folded monomers (*Newcomer et al., 2018*). The high quality of the NMR data enabled us to obtain nearly complete (98%) NMR assignments for the monomeric Dec protein (*Newcomer et al., 2018*). The NMR structure was determined using the program ARIA 2.3 (*Bardiaux et al., 2012*), based on the experimental NMR constraints summarized in *Table 2*. The coordinates of the Dec monomer NMR structure ensemble have been deposited to the PDB under accession number 6E3C.

The N-terminal domain of the Dec monomer has an OB-fold motif (*Figure 2*). The final NMR structure (*Figure 2A*) is close to an initial NMR model calculated with the program CS-ROSETTA (*Shen et al., 2009*), based on the assigned NMR chemical shifts in combination with template-based modeling (*Figure 3—figure supplement 2*). The backbone (Cα, C', N, O) RMSD is 2.3 Å over the folded parts of the structure (residues 12–89) between the NMR structure and the CS-Rosetta model. Submission of the NMR Dec monomer structure to the DALI server (*Holm and Rosenström, 2010*) identified about 80 matches to proteins that contain similar OB-fold domain structures. The strongest of these is a portion (residues 117–174) of chain E of the 40S ribosomal protein SA (PDB code 4KZX-E). This superposed on the Dec NMR structure with a Cα RMSD of 3.5 Å over 61 residues, supporting our interpretation that the structure of the Dec N-terminus is in the family of OB-fold motifs.

The canonical OB-fold motif (*Murzin, 1993*) is a five-stranded Greek key β-barrel, comprised of a β1-β3 meander and a β4-β5 hairpin, with an α-helix 'αOB' intervening between strands β3 and β4. The five-stranded OB-fold β-barrel is closed by an antiparallel pairing between strand β1 and β4, and a short parallel pairing between stands β3 and β5 (*Murzin, 1993*). In the Dec structure the antiparallel pairing of stands β1 and β4 is conserved, but strands β3 and β5 are too distant (~13 Å) for any H-bond contacts. Thus, the Dec β-sheet is distorted to a more open structure compared with the classical five-stranded OB-fold β-barrel (*Figure 2B*). In general, the cores of OB-fold β-barrels are consolidated by three layers of hydrophobic residues (*Murzin, 1993*; *Alexandrescu et al., 1995*). This arrangement is also present in the Dec structure (not shown). The canonical role of the helix αOB is to provide a 'hydrophobic plug' for the bottom hydrophobic layer of the β-barrel (*Murzin, 1993*; *Alexandrescu et al., 1995*). Residue V55 appears to serve this role in Dec. In OB-fold proteins the orientation of the helix αOB is more variable than that of the β-barrel (*Alexandrescu and Shortle, 1994*; *Guardino et al., 2009*), and in Dec, the αOB helix extends almost directly between strand β3 and β5, with the helix axis in the plane of the β1-β3 meander, rather than below this structure (*Figure 2B*).

As with many OB-fold proteins (*Guardino et al., 2009*), Dec has additional non-conserved elements of secondary structure at the periphery of the conserved motif. The short strand β0 forms an anti-parallel pair with the N-terminus of the OB-fold strand β1, and the helix α2 extends away from

**Table 2.** Statistics for the top 20 NMR structures of Dec.

**Experimental restraints**

| | | |
|---|---|---|
| Total number of NMR restraints | 1009 | |
| Total number of NOE distance restraints | 767 | |
| Ambiguous | 73 | |
| Unambiguous | 694 | |
| Long range ($|i-j| > 4$) | 124 | |
| Medium range ($|i-j| \leq 4$) | 93 | |
| Sequential ($|i-j|=1$) | 281 | |
| Intra-residue NOEs | 269 | |
| Total number of dihedral restraints | 174 | |
| $\varphi/\ \psi$ | 127 | |
| $\chi_1$ | 47 | |
| Hydrogen bond restraints (34*2) | 68 | |
| *RMSD from experimental restraints*[*] | | |
| NOE distance (Å) | 0.054 ± 0.003 | |
| Dihedral (°) | 0.42 ± 0.14 | |
| *RMSD from ideal geometry* | | |
| Bonds (Å) | 0.0042 ± 0.0001 | |
| Angles (°) | 0.55 ± 0.02 | |
| Improper angles (°) | 1.68 ± 0.12 | |
| $E_{L-J}$ (kcal/mol) | - 316 ± 44 | |
| *RMSD from mean NMR structure* | *Backbone*[†] | *All Heavy Atoms* |
| Entire protein 1–134 (Å) | >9.5 | >9.5 |
| Folded regions 12–86 (Å) | 1.33 ± 0.21 | 1.95 ± 0.32 |
| OB-fold 18–77 (Å) | 1.08 ± 0.18 | 1.66 ± 0.25 |
| OB-fold 2° structure[‡] (Å) | 0.84 ± 0.15 | 1.57 ± 0.31 |
| *Procheck Ramachandran Plot Statistics*[§] | | |
| Most favored (%) | 89.1 | |
| Additionally allowed (%) | 10.9 | |
| Generously allowed (%) | 0.0 | |
| Disallowed (%) | 0.0 | |
| *Quality Z Scores from PSVS*[e] | | |
| Procheck ($\phi,\psi$) | −2.64 | |
| Molprobity Clash | −1.19 | |

[*] Structures had no NOE violations > 0.5 Å nor dihedral violations > 5 degrees.

[†] Atoms: Cα, N, C, O.

[‡] Calculated over residues in the OB-fold portion: 21–32 (β1), 35–40 (β2), 47–50 (β3), 52–59 ($\alpha_{OB}$), 63–68 (β4), 73–77 (β5), a total of 41 amino acids.

[§] Calculated with the PSVS server (http://psvs-1_5-dev.nesg.org) using only the folded parts of Dec (residues 12–89).

DOI: https://doi.org/10.7554/eLife.45345.007

the last OB-fold strand β5 (*Figure 2C*). Outside of the folded globular part of the Dec monomer structure, residues 1–11 and 90–134 are disordered (*Figure 2B*). Although these segments were included in the NMR structure calculations, they have no defined structure in the Dec monomers (*Figure 2B*).

To further characterize Dec, we analyzed the dynamics of the monomeric protein using NMR [15]N-relaxation data in terms of the 'Model-Free' formalism (*Figure 3—figure supplement 1*). The local

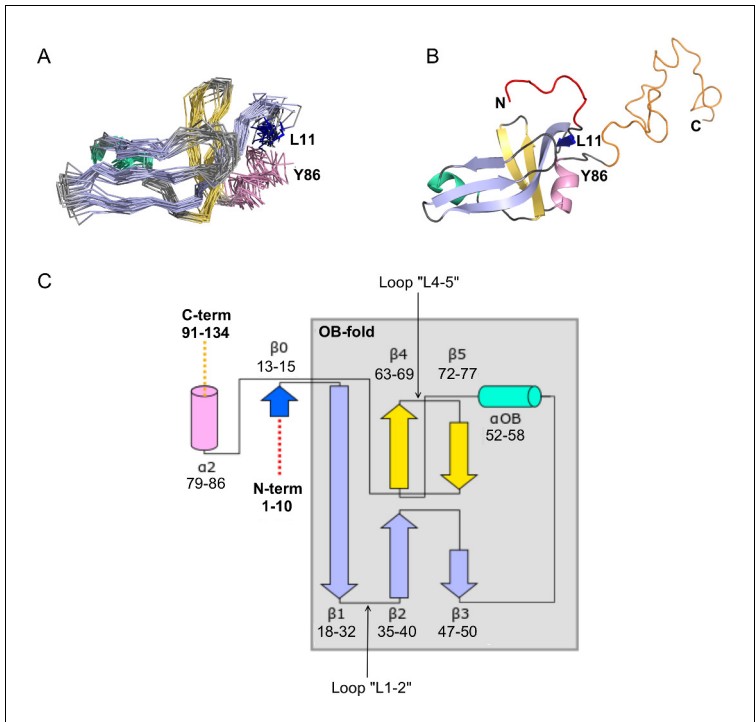

**Figure 2.** NMR structure of the Dec monomer. (**A**) Ensemble of the 20 lowest-energy NMR structures. For clarity, the disordered N- and C-termini are not shown. The folded globular part of Dec has an OB-fold consisting of a β1-β3 meander (sky blue) and a β4-β5 hairpin (yellow), with an α-helix (green) intervening between strands β3 and β4. Additional secondary structure outside of the OB-fold incudes the short N-terminal strand β0 (dark blue), and a C-terminal α-helix (pink). (**B**) Ribbon diagram showing the structure closest to the NMR average. The first 10 N-terminal residues and the last ~45 C-terminal residues, which are unstructured in the Dec monomers are colored red and orange, respectively. The coloring scheme for the protein in this and subsequent panels is the same as in A. (**C**) Diagram summarizing the topology and secondary structure limits of the Dec monomers.
DOI: https://doi.org/10.7554/eLife.45345.008

backbone mobility of the Dec monomers is summarized in *Figure 3*. The N-terminus from residues 1–11 and the C-terminus from residues 90–134 have small $S^2$ order parameters characteristic of unfolded protein segments (*Figure 3A*, *Video 2*). The random coil chemical shifts of these regions (*Newcomer et al., 2018*) and the lack of long-range NOEs, are also consistent with these segments being unstructured in the Dec monomers. Small and intermediate $S^2$ values indicative of increased flexibility on the ps-ns timescales are also seen at sites within the folded globular portion of Dec (*Figure 3A*). Information on motions within slower µs-ms timescales can be garnered from exchange contributions to R2 transverse relaxation (R2$_{EX}$ parameters). The sites with the largest R2$_{EX}$ contributions are similar to those with low S2 order parameters, in particular loops L23, L45 and the region surrounding the loop L3α (purple and red in *Figure 3B*). The sites within the folded portion of the Dec monomers with the largest flexibility based on $S^2$ and R2$_{EX}$ values are shown in *Figure 3C and D*, respectively.

## Homology modeling of phage-bound Dec suggests a trimeric β-helix motif forms the C-terminal spike

The NMR structure of Dec revealed an OB-fold for the N-terminal portion of the protein; however, the C-terminal portion was unstructured for the monomer in solution. As we previously observed (*Parent et al., 2012b*), the N-terminal domain forms the legs of the Dec 'tripod' that are bound to the capsid, and the C-terminal region forms its protruding central stalk. The ordered N-terminal domain was not sufficient to fill the entire density within the cryo-EM map. Moreover, fitting the structured N-terminal OB-fold region of the protein into the density showed little or no inter-subunit trimer contacts from the first 77 residues of Dec. Thus, the C-terminus of Dec, evident as a spike in

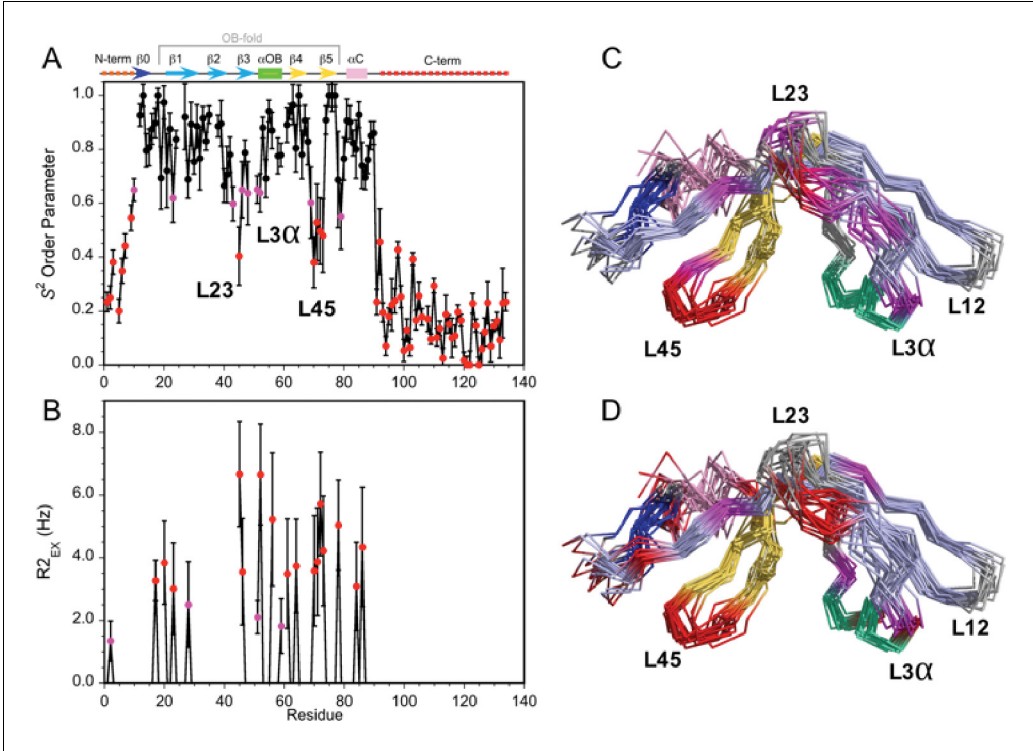

**Figure 3.** Dynamics of the Dec monomer. (**A**) $S^2$ order parameters describing the amplitude of fast motions on the ps-ns timescale. Rigid ($S^2 > 0.65$), moderately flexible ($0.65 \geq S^2 > 0.55$), and highly flexible ($0.55 \geq S^2$) sites are indicated by black, purple, and red symbols respectively. The secondary structure of Dec is given at the top of panel A. (**B**) Contributions to R2 relaxation from slow conformational exchange on the μs-ms timescale. Amide protons with moderate $R2_{ex}$ values smaller than 3 Hz are shown in purple, those with large contributions above 3 Hz in red. The $S^2$ and $R2_{ex}$ parameters were obtained from a Model-Free analysis (*Lipari and Szabo, 1982*) of $^{15}N$ R1, R2, and $^{1}H$-$^{15}N$ NOE relaxation data for Dec (*Figure 3—figure supplement 1*) using the program TENSOR 2.0 (*Dosset et al., 2000*). In (**C**) and (**D**) the $S^2$ and $R2_{ex}$ values are mapped onto the NMR structure ensembles for Dec. Residues 1–10 and 90–134 which are unfolded and thus have the lowest $S^2$ values, are not shown in the structures.

DOI: https://doi.org/10.7554/eLife.45345.009

The following figure supplements are available for figure 3:

**Figure supplement 1.** $^{15}N$ relaxation values for monomeric Dec.

DOI: https://doi.org/10.7554/eLife.45345.010

**Figure supplement 2.** CS-ROSETTA modeling of the Dec monomer structure.

DOI: https://doi.org/10.7554/eLife.45345.011

cryo-EM maps of capsid-bound Dec (*Parent et al., 2012b*), becomes ordered either upon Dec trimerization or capsid binding.

We used a combination of the NMR structure and homology modeling to build a model of the capsid-bound Dec trimer using the cryo-EM density map as a guide. We initially docked the NMR-derived OB-fold portion of the Dec structure (residues 10–77) into the segmented cryo-EM density, independently fitting three copies of the OB-fold. Using Phenix, each copy was refined by flexible fitting to accommodate the slight differences in the asymmetric trimer density. We then built a complete trimer by using a combination of homology modeling and flexible fitting to refine the C-terminal portion of Dec separately from that of the OB-fold. Sequence homology of the C-terminus most closely matched a fragment from the bacteriophage T4 trimeric proximal long tail fiber protein gp34 (PDB ID: 4UXE; *Granell et al., 2017*) (see Materials and methods for details), and was fit to the segmented cryo-EM density. In brief, the C-terminal domain (residues 78–134) was modeled as a three-stranded β-helix domain based on the remote homology model, consistent with

predicted secondary structures of the tail fiber protein gp34, and to fit the cylindrical shape of the cryo-EM density for the C-terminal domain. The C-terminus of the final trimeric Dec structure was then further refined using partially flexible fitting preserving the β-helix domain as a rigid unit within the cryo-EM density envelope (*Figure 4*). These residues are predicted with a lower confidence owing to the flexibility and lower resolution within that portion of the structure and were refined using the Cα backbone only in this region.

## Mutagenesis of P22 coat protein and Dec reveals key residues that modulate binding

The OB-fold of Dec, as determined by NMR, fits the cryo-EM map of the 'tripod' legs in the native phage L virion very well (*Figure 4*). This allowed us to fit models of several coat protein subunits and Dec to analyze the Dec-coat protein binding interface. The resolution of the coat protein in the cryo-EM map in this region was sufficient to fit some bulky side chains. Furthermore, the NMR structure allowed us to make predictions regarding the side chain orientation of Dec residues that potentially bind to coat protein. Analysis of the binding interface revealed several residues in close contact between the phage L coat protein and Dec that appear to be important for binding. Therefore, we tested our models using site-directed mutagenesis of phage P22 coat protein. P22 is a very well-established model system with a variety of molecular tools readily available (*Teschke and Parent, 2010*). Previous work has shown that P22 and phage L behave similarly (*Gilcrease et al., 2005*; *Tang et al., 2006*; *Parent et al., 2012b*) and all the amino acid substitutions found in phage L have even been identified occasionally in P22 stocks as phenotypically silent mutations (Teschke lab, unpublished data). Lastly, as Dec bound to P22 is a widely used system for a variety of nanomaterials

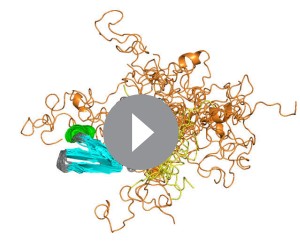

**Video 2.** Definition of different regions of the Dec monomer NMR structure. To illustrate the precision of the structure all 20 members of the NMR ensemble are shown. The movie then flips through the 20 individual NMR structures to show the differences in precision between the structured OB-fold component, and the unstructured N- (yellow) and C-termini (orange). In the second half of the movie, the N- and C-termini are not shown to better illustrate differences in structural definition of the OB-fold component.
DOI: https://doi.org/10.7554/eLife.45345.012

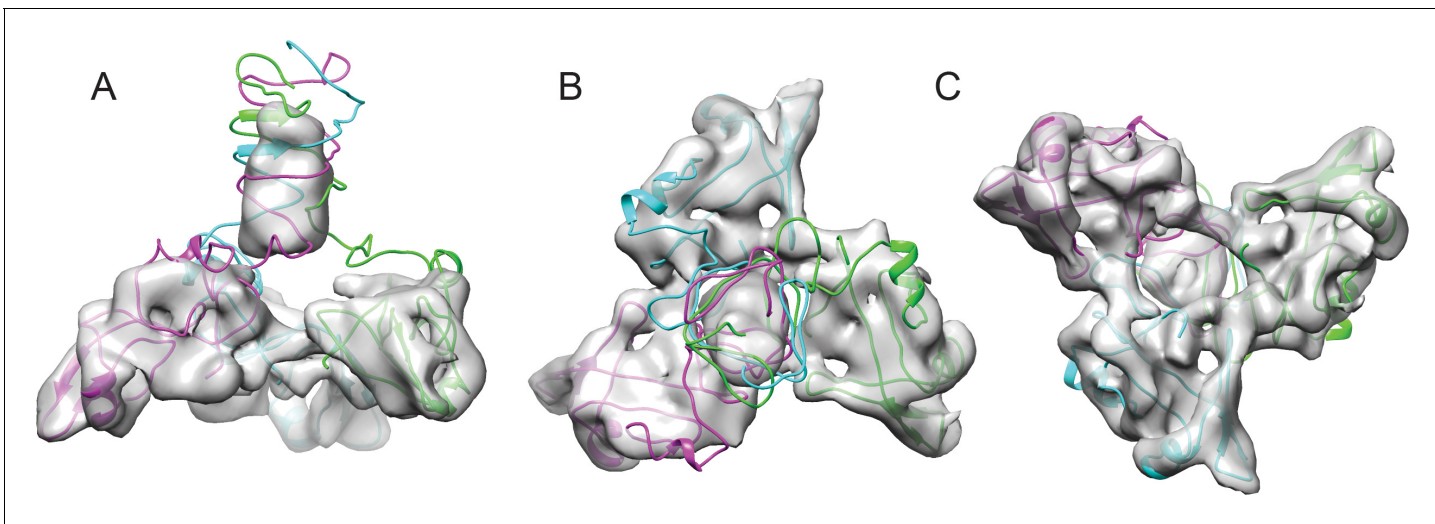

**Figure 4.** Fitting the NMR OB-fold into the cryo-EM map and model of the trimer. (**A**) Side B) top-down, and C) bottom-up views of the Dec trimer model fit into the cryo-EM density map with individual chains colored magenta, cyan, and green.
DOI: https://doi.org/10.7554/eLife.45345.013

applications (*Sharma et al., 2017*; *Schwarz et al., 2015*; *McCoy et al., 2018*), identifying key residues that contribute to Dec interactions with P22 is highly useful for practical applications. Therefore, we felt it reasonable to use the established genetics of P22 as a model for the native phage L for our mutagenesis studies.

Previous work suggested that Dec binding to P22 capsids is driven primarily through electrostatic interactions (*Schwarz et al., 2015*), and our data also suggest a charged binding interface. To assess the role of specific coat protein surface residues in modulating Dec binding, the following five coat protein residues in potential close contact with Dec were chosen for single amino acid substitution by site-directed mutagenesis: E81R, P82S, R299E, P322S, and E323R (*Figure 5*; *Video 3*). Site-directed mutagenesis was performed on a plasmid that expresses coat protein (see Materials and methods) as described previously (*Parent et al., 2007*; *Suhanovsky et al., 2010*; *Suhanovsky and Teschke, 2013*). Cells carrying the plasmid were infected with phages having a nonsense mutation in gene 5 (encodes coat protein), so that any phages produced are the result of complementation by the coat gene expressed from the plasmid. None of the amino acid substitutions affected virion production or infectivity; phages assembled with each coat protein variant were grown to high titer and were phenotypically indistinguishable from the parental phage.

Coat proteins occupying different local conformations (as a result of the quasi-equivalent capsid lattice) contribute different residues to the binding interface (*Figure 5A*, *Video 3*). For example, residues E81, P82, and R299 from the coat protein subunits that form the quasi-three-fold symmetry axes (grey subunits in *Figure 5*) contact Dec from one subunit, whereas residues P322 and E323 contact Dec from adjacent and overlapping coat protein subunits (black subunits in *Figure 5*). To assess the role that specific Dec residues play in capsid binding, we also made five single amino acid substitutions in recombinant Dec protein: K30D, Y31A, Y49E, Y71A, and E73R (see Materials and methods). Modified Dec proteins were added to mature WT phage particles, complexes were purified using CsCl gradients, and relative amount of each Dec protein bound to virions was quantitated. The global secondary structure of all Dec variants was indistinguishable from the WT protein by circular dichroism. Furthermore, native gel experiments performed as described previously (*Newcomer et al., 2018*) showed that all Dec variants assemble as trimers in solution. Therefore, any changes in Dec occupancy in the variants are likely due to a disruption of the binding interface, rather than protein folding defects.

Among the five coat protein changes, coat amino acid substitutions E81R and E323R attenuated but did not completely abolish Dec-binding ability (*Figure 5B,C*). We created a variant containing both amino acid substitutions, E81R:E323R, which displayed less Dec binding (<20%) than each of the individual substitutions. Since the E81R:E323 double mutant affects both Dec-binding sites, our data indicate that the binding interface involves more than these two residues. It is also possible that given the inherent asymmetry in the capsid quasi three-fold binding sites for Dec because they are formed by coat proteins in two distinct quasi-equivalent conformations, not all chains of the Dec trimer have equal effects form the amino acid substitutions. Dec with amino acid substitutions K30D and Y49E displayed lower binding than the WT protein, consistent with the prediction of an electrostatic binding interface (*Figure 5D*, *Video 3*).

The close contact region between Dec and coat protein can be thought of as including two binding sites. One site includes coat residues E81 and P82 that interact with Dec residues K30, Y31 and Y49 (*Figure 5D*, 'site 1'). Dec residue Y49 is directed toward the side chain of coat residue E81. The other site includes Dec residues Y71 and E73 and coat residues P322, E323 and R299 (*Figure 5D*, 'site 2'). Overall, variations in site 1 had the largest effect on binding. In Dec, site 1 residues K30 and Y49 had the largest effect on binding saturation, and in coat protein site 1 residue E81R had the largest effect relative to the other substitutions. Although site 1 clearly plays a strong role in modulating Dec binding, coat residue E323R that occupies site 2 had a modest but reproducible effect. We conclude that the site 1 plays a larger role in mediating Dec binding to coat protein, and is largely due to electrostatic interactions.

We also tested our hypothesis by making substitutions in coat protein residues that contact Dec near, but not within sites 1 and 2. As expected, these did not reduce binding (namely coat residues P82S, R299E and P322S; we intentionally made conservative substitutions for prolines as coat protein is highly aggregation prone; *Teschke and Parent, 2010*). Similarly changes in Dec near, but not within sites 1 and 2, also did not reduce binding, including Y31A, Y71A, and E73R. These

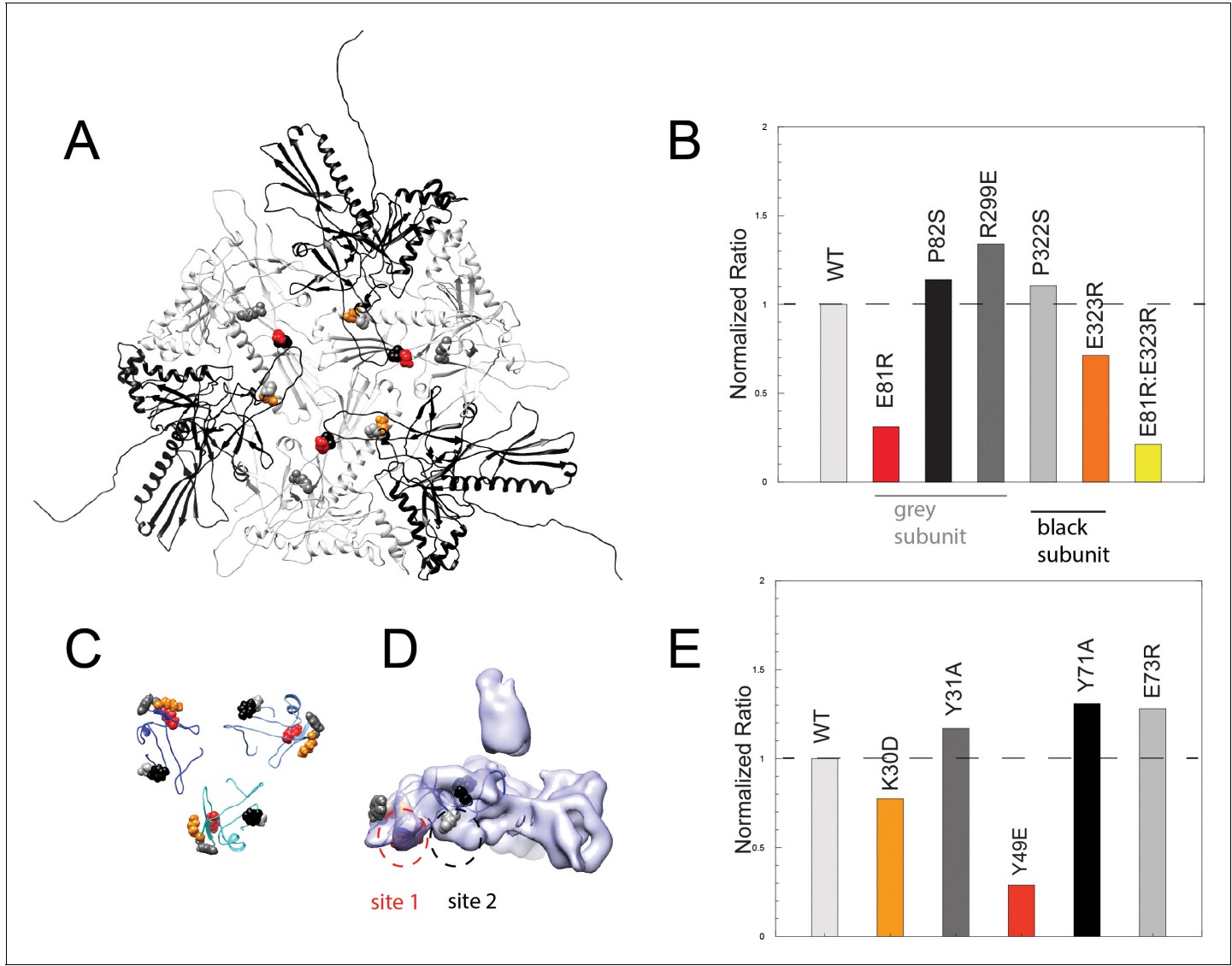

**Figure 5.** Binding assays of variant coat proteins and Dec to probe the interaction interface. (**A**) Coat protein subunits depicted as ribbon diagrams around a quasi three-fold axis Dec-binding site. Three coat subunits directly surrounding a quasi three-fold axis are shown in light grey, and three neighboring coat proteins are shown in black. Residues selected for mutagenesis are shown as spheres. (**B**) Normalized binding data of the ratio of variant coat protein bound to WT Dec, color-coded to match the corresponding residues in panel A. (**C**) Portions of Dec highlighting the OB-fold (residues 10–77) shown as ribbons, with each monomer a different shade of blue. Residues selected for mutagenesis are shown as spheres. (**D**) Enlarged side view of the Dec cryo-EM density with the OB-fold (residues 10–77) from one monomer shown as a ribbon, with the two different capsid-binding sites indicated. (**E**) Normalized binding data of the ratio of WT coat protein bound to variant Dec, and color-coded to match the corresponding residues in panels C and D.

DOI: https://doi.org/10.7554/eLife.45345.014

observations are consistent in light of the 3D capsid structure, and indicates that sites 1 and 2 are the true binding sites.

## Discussion

### Decoration proteins employ a variety of binding schemes to adhere to phage capsids

For dsDNA tailed phages and related viruses, decoration proteins can occupy the capsid lattice in several different positions. Some decoration proteins bind capsids at the center of hexamers

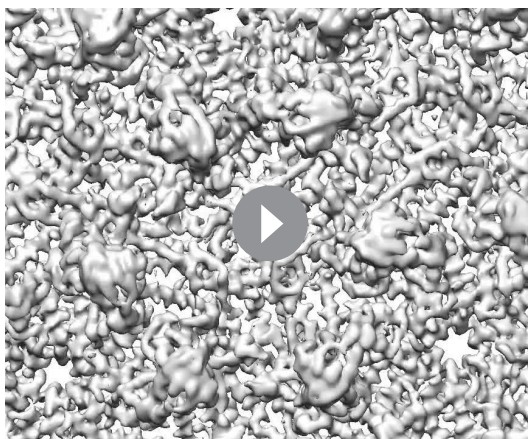

**Video 3.** Key residues that modulate Dec-binding affinity. View of the binding interactions between Dec and coat protein at a quasi-three-fold symmetry axis. Coat protein is shown in grey density and Dec is shown in lavender. Six coat subunits are fit within this density and three Dec OB-fold domains from the NMR structure were fit to the Dec density. Individual protein chains are shown are ribbons; amino acid sites that were chosen for mutagenesis are shown as spheres, and color coded to match the data in *Figure 5* of the main text.

DOI: https://doi.org/10.7554/eLife.45345.015

including those of phages T4, T5, RB49, and Sf13 through Sf19, (*Vernhes et al., 2017*; *Doore et al., 2018*; *Sathaliyawala et al., 2010*; *Fokine et al., 2011*). Other decoration proteins bind the edges of hexamers as exemplified by gp17 in phage N4 (*Choi et al., 2008*). Finally, there are decoration proteins that bind the capsid in areas between capsomers such as gpD in λ, gp56 in phage TW1, Soc in phage T4, gp87 in phage P74-26, and Dec in phage L (*Lander et al., 2008*; *Tang et al., 2006*; *Wang et al., 2018*; *Qin et al., 2010*; *Stone et al., 2018*). In all the aforementioned cases, the decoration proteins are added after capsid assembly. Conversely, the herpesvirus heterotrimeric triplex protein decorates the outside of the capsid and is also essential during capsid assembly (*Zhou et al., 1994*; *Heming et al., 2017*). For the majority of these examples, all possible quasi-equivalent binding sites on the lattice are occupied by the decoration proteins. By contrast, in phage L, Dec uniquely binds at one type of symmetry axis with high affinity: the quasi-three-fold sites between hexamers (*Figure 1A*).

To function in a stabilizing capacity, decoration proteins must bind capsids with high enough affinity to remain associated even under harsh environmental conditions. As described above, there are many potential binding sites on capsids, so the requirement for Dec binding affinity must be balanced against binding specificity, to allow discrimination between binding sites. Indeed, there is a wide range of reported $K_D$s for various decoration proteins. Phage L's Dec and T4's Hoc bind their respective capsids with nM affinity, whereas pb10 binds phage T5 with pM affinity (*Parent et al., 2012b*; *Vernhes et al., 2017*; *Shivachandra et al., 2006*). Taken together, variations in binding affinities and saturating versus discriminating binding behavior suggests that decoration proteins are capable of recognizing subtle differences in capsid lattices. The reason some decoration proteins evolved to discriminate between similar quasi-equivalent sites is unknown, but at some positions on the capsid lattice coat-coat interactions may be stronger than at other quasi-equivalent positions; therefore, some positions may not require stabilization from a decoration protein. Yet, other positions may be more vulnerable to environmental assaults than others, and benefit from additional stabilization.

How might Dec discriminate between the different three-fold sites on phage L? *Figure 6* shows a comparison of the coat protein subunits surrounding two types of quasi three-fold axes. Coat residues that interact with Dec, in particular E81 (see above), shows a substantial difference in position at these different capsid locations. At the quasi three-fold site between hexons, coat protein residue E81 clearly interacts with Dec binding site 1 at all three subunits in Dec. However, at the quasi three-fold site between hexons and pentons, the environments surrounding E81 are spread further away from the Dec-binding site: if one coat monomer were aligned to a Dec monomer, E81 would be 7.3 and 10.4 Å away in the other two coat protein chains, respectively, which is too great a distance to allow binding. By contrast, at the icosahedral three-fold binding interface, the E81 residue is within 4 Å of the Dec binding site 1 for two out of three monomers. Only E81 in the third subunit is significantly distant from the Dec binding site 1 (8.8Å away). These coat protein topology differences are likely responsible for the failure of Dec to bind to the type of quasi three-fold site surrounding pentamers and the different Dec-binding affinity from the icosahedral three-fold site as measured in vitro (*Parent et al., 2012b*). The coat protein subunits surrounding the quasi three-fold symmetry axes between pentamers and hexamers (green circles in *Figure 1A*) that do not bind Dec have the

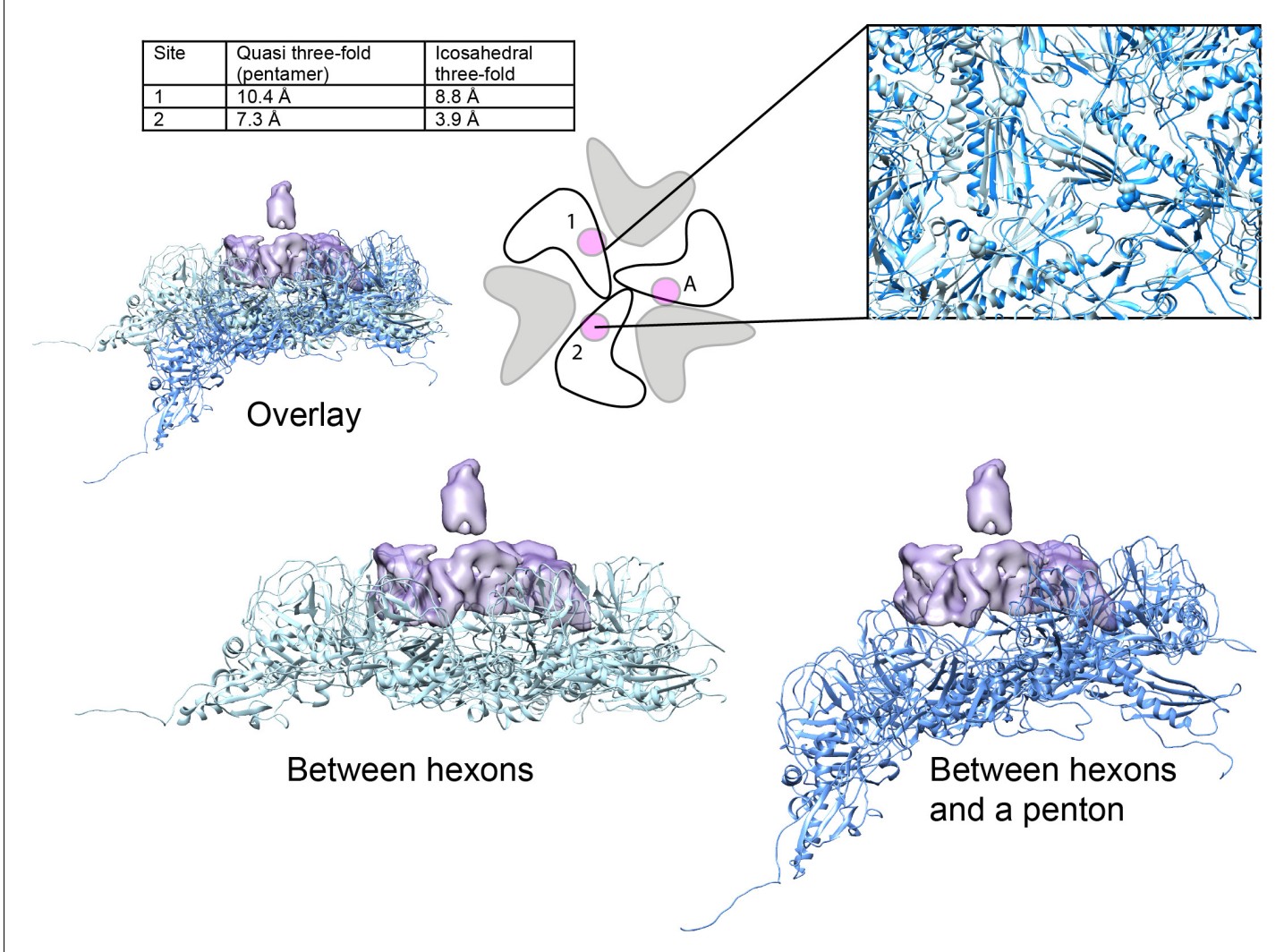

| Site | Quasi three-fold (pentamer) | Icosahedral three-fold |
|------|------------------------------|------------------------|
| 1    | 10.4 Å                       | 8.8 Å                  |
| 2    | 7.3 Å                        | 3.9 Å                  |

**Figure 6.** Comparison of Dec-binding residues at various three-fold symmetry sites on the phage L capsid. The topology of six coat protein subunits surrounding the different three-fold positions were compared using the Match tool in Chimera. The quasi three-fold site between coat protein hexamers that binds a Dec trimer was chosen as the reference structure and aligned to the three-fold site between a pentamer and neighboring hexamers (dark blue) that does not bind Dec. The cartoon schematic shows the location of residue E81 with a small magenta circle on three coat protein subunits that comprise the Dec binding interface (label 'A' designates the specific coat protein subunit used for anchoring the structures in Chimera). The relative displacement of E81 in subunits labeled '1' or '2' when in an icosahedral three-fold or the quasi three-fold surrounding pentamers was measured and the distances are shown in the table. A side view of the quasi three-fold site between hexamers that binds Dec (light blue) with the quasi three-fold site between a penatmer and neighboring hexamers (dark blue) is shown as a ribbon diagram. The box shows an enlarged and tilted view of the binding interface with residue E81 shown as spheres. Additionally, the Dec density is shown in lavender at the position of the quasi three-fold site between hexamers.

DOI: https://doi.org/10.7554/eLife.45345.016

greatest surface curvature with highly bent contacts between capsomers (*Figure 6*). These interfaces are a weak point in many capsids as perturbations such as chemical or thermal treatments can cause entire pentamers to be released from the capsid (*Parent et al., 2012b*; *Teschke et al., 2003*). The quasi three-fold coat protein subunits between hexamers that do bind Dec also have some curvature, but it is less extreme than the quasi three-folds between pentamers and hexamers (*Figure 6*), likely why Dec can bind here, albeit at orders of magnitude lower affinity (*Parent et al., 2012b*). By contrast, the icosahedral three-folds sites are the flattest, and although some capsids use decoration proteins to stabilize this interface, Dec is not used this way in native phage L particles, indicating a potential difference in the role of Dec compared to other decoration proteins. Biophysical

measurements have shown that Dec plays a much larger role in stabilizing dsDNA-containing heads compared to empty capsids, and atomic force microscopy experiments showed Dec indeed binds capsids at positions most prone to mechanical damage (i.e. quasi three-fold sites between hexons) (*Llauró et al., 2016*). Therefore, Dec binds specifically at the quasi three-fold axes between hexamers via specific residues within the key binding site (Dec site 1), and displacement of the corresponding binding partner residues within the capsid lattice decreases binding affinity at other capsid locations. Although Dec has been shown to stabilize capsids, perhaps this is not its only function (see further discussion below).

## Dec has a novel architecture

In dsDNA-containing tailed phages, previously studied decoration proteins for which atomic-resolution structures are known have three basic types of protein folds. Decoration proteins such as phage T5's pb10 and T4's hoc, have an overall Ig domain fold (*Vernhes et al., 2017*; *Fokine et al., 2011*). A second, more common type of decoration protein, exemplified by phage lambda's gpD, P74-26's gp87, phage 21's SHP, and TW1 gp56, have similar polypeptide folds and form a symmetric trimer with an N-terminal β-tulip domain and an α/β subdomain that binds and stabilizes capsids through hydrophobic interfaces (*Stone et al., 2018*; *Lambert et al., 2017*). Additionally, some decoration proteins are known to bind the center of hexamers, as exemplified by protein Psu in phage P4 (*Isaksen et al., 1993*; *Dokland et al., 1993*; *Banerjee et al., 2012*). Our work shows that Dec represents a fourth type of decoration protein fold (*Figure 7*), the OB-fold, as well as a different capsid-binding mechanism that discriminates between quasi three-fold capsid-binding sites between pentamers and hexamers, and the icosahedral three-fold sites. To our knowledge, Dec represents the first occurrence of an OB-fold structure in a virus decoration/cementing protein. Dec may have additional functions besides capsid stabilization. For example, Dec could play a role in cell adhesion. Preliminary evidence suggests that Dec might bind to target cells, presumably through carbohydrate moieties emanating from the target cell surfaces (Teschke lab, unpublished data). From a structural point of view, both the OB-fold part of Dec and the C-terminal beta-helix could be carbohydrate-binding motifs. However, future work will be needed to fully explore and validate this hypothesis. If Dec plays a role in cell surface binding, 60 Dec trimers rather than the saturating 140 must be sufficient for this function.

## Dec binds phage L capsids with an unusual strategy

Crystal structures for gpD (*Yang et al., 2000*), gp87 (*Stone et al., 2018*), and SHP (*Forrer et al., 2004*) lack resolution for several N-terminal residues, indicating that these regions are flexible when these decoration proteins are not associated with capsids. Cryo-EM maps of gpD in native phage λ show that the N-terminus becomes highly ordered when gpD is in a bound conformation, and that the N-terminus is the major part of the capsid-binding mechanism as the N-terminus forms a stabilizing β-sheet with strands supplied from the capsid protein (*Lander et al., 2008*). Like gpD, the N-terminus of the Dec monomer is disordered in solution as shown here by NMR. However, unlike gpD, our data indicates that the Dec N-terminus remains rather flexible in the capsid-bound form since there is (1) no cryo-EM density that is attributable to the first 10 residues, (2) our previous cryo-EM data show the N-terminus of Dec can be labeled with large cargo such as nanogold beads that are rather flexibly bound to the capsid (*Parent et al., 2012b*), and (3) deletion of the first 11 residues of Dec does not affect capsid binding affinity (*Schwarz et al., 2015*). When we compare the phage L Dec binding motif to the capsid stabilization mechanism in HK97, we see that the residues that control Dec binding occupy similar spatial positions as the HK97 crosslinking residues that crosslink to form the lattice 'chainmail', suggesting that stabilization at this type of interface is crucial to capsid stability.

Furthermore, comparison of Dec to the β-tulip family of decoration proteins highlights some interesting implications about binding affinity and selectivity of binding locations in terms of symmetry of the auxiliary proteins involved. Perfect trimeric symmetry in a bound decoration protein may allow a broader binding specificity. For example, gpD and gp87 are highly symmetric trimers and these decoration proteins occupy all quasi-equivalent binding sites including icosahedral three fold symmetry axes and also both types of quasi-three-fold sites. By contrast, the asymmetry in the bound Dec trimer evinced by cryo-EM is an unusual property among decoration proteins and

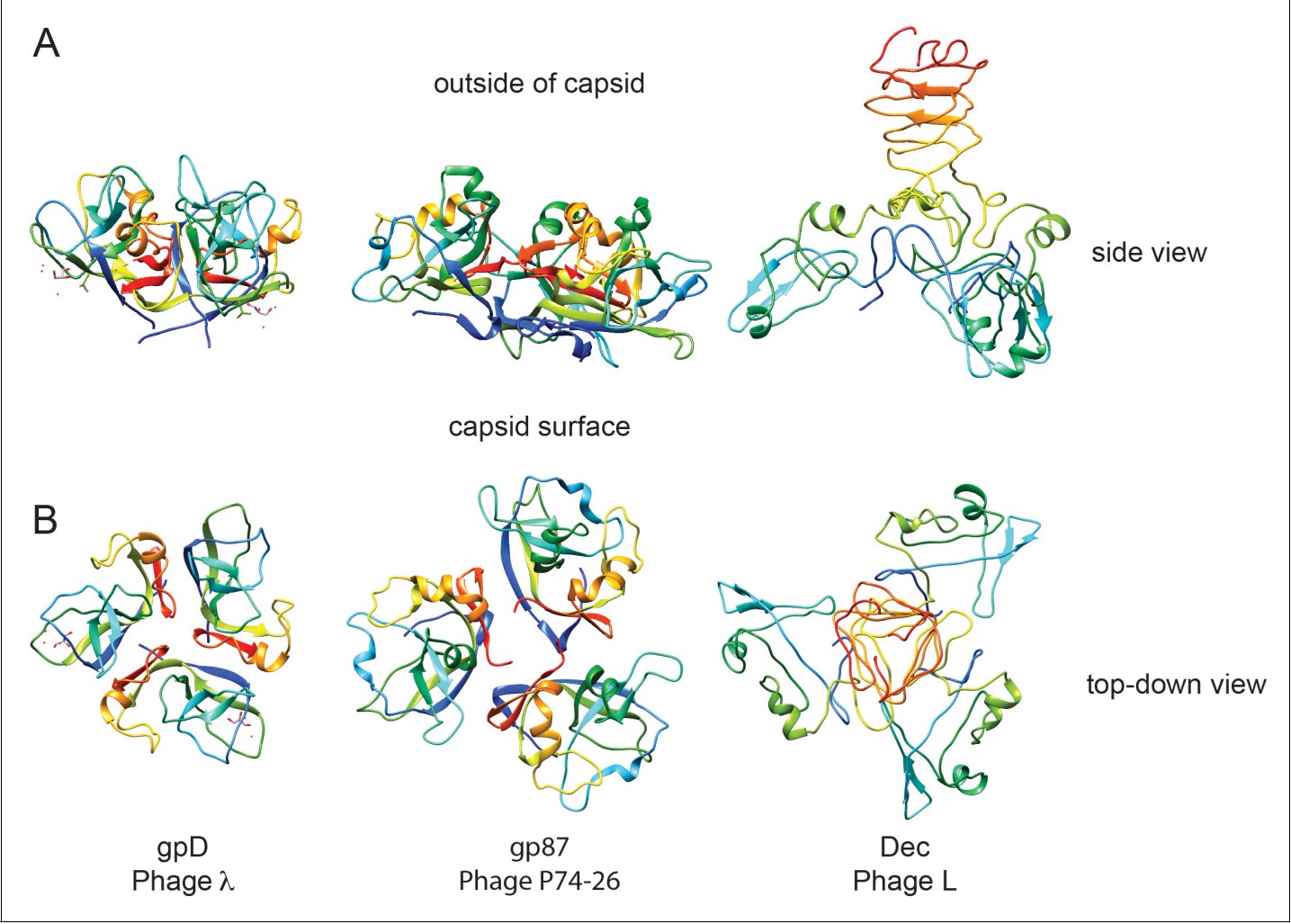

**Figure 7.** Comparison of decoration proteins. (**A**) Side and **B**) top-down views of decoration proteins gpD (PDB ID 1C5E), gp87 (PDB ID 6BL5) and Dec (PDB ID 6D2D). Chains of each trimer are rainbow colored from N-terminus (blue) to C-terminus (red).
DOI: https://doi.org/10.7554/eLife.45345.017

correlates with its binding selectivity to only the quasi three-fold sites between hexamers. Dec appears to represent a separate evolutionary lineage of decoration proteins that is distinctly different than those in the β-tulip family and whose capsid binding is similarly driven mainly by interactions in site 1, with a smaller contribution towards binding in site 2.

## Materials and methods

**Key resources table**

| Reagent type (species) or resource | Designation | Source or reference | Identifiers | Additional information |
|---|---|---|---|---|
| Strain, strain background (*Salmonella enterica*) | DB7136 (*Salmonella enterica* serovar Typhimurium; used to purify phages) | PMID: 15743953 | | Host for phages P22 and L |

*Continued on next page*

*Continued*

| Reagent type (species) or resource | Designation | Source or reference | Identifiers | Additional information |
|---|---|---|---|---|
| Strain, strain background (*Escherichia coli*) | BL21/DE3/pLysS (*Escherichia coli*; used for protein purification) | Invitrogen | | Protein expression system |
| Strain, strain background (Phage P22) | Phage P22 (P22 5-am N114) | PMID: 4773026 | | Phage, clear plaque mutant |
| Strain, strain background (Phage L) | Phage L | PMID: 15743953 | | Phage, clear plaque mutant |
| Recombinant DNA reagent | pMS11 (plasmid-coat protein; used in binding assays) | PMID: 24126914 | | Plasmid for coat protein complementation |
| Recombinant DNA reagent | pDec (plasmid-Dec protein; used in binding assays) | PMID: 22575828 | | Plasmid for Dec protein purification-C-terminal his tag |
| Recombinant DNA reagent | pDec-NMR (plasmid-Dec protein; used in NMR experiments) | PMID: 30109462 | | |
| Sequence-based reagent | Primers-coat protein mutants: E81R (gtaaacatgggaccgccgg ataacgacttcttccagttgcg); P82S (gcggtaaacatgggagag tccgataacgacttcttccagttgcg); R299E (gcgactttctccgtagtcg aagttgttgacggtactcatgttg); P322S (gatgtttccctgtcttccgag cagcgtgcctacgccaacgtt); E323R (gatgtttccctgtctccgcgc cagcgtgcctacgccaacgtt) | IDT | | |
| Sequence-based reagent | Primers-Dec protein mutants: K30D (gtgtctgcgcttccgattaaag ctatcgagtacgctaatgacgg); Y31A (tgtctgcgcttccgattaaagc tatcaaagccgctaatgacgga); Y49E (ggccccgtatgctgaccagg agatgtcagcgcaaacagtagcc); Y71A (ggatatctgttccggagccagg ccggcgagctgctctatatgagc); E73R (ggatatctgttccggagccag tacggcaggctgctctatatgagc) | IDT | | |
| Commercial assay or kit | Quikchange | Agilent | | |
| Commercial assay or kit | Mini prep kits | Qiagen | | |
| Compound, chemical or drug | Isopropyl β-D-1-thiogalac topyranoside | GoldBio | | Protein expression |
| Compound, chemical or drug | Ampicillin | Amresco | | Selection media |
| Other | LB (broth and agar) | Invitrogen | | Media |
| Other | 15N- Ammonium Chloride | Cambridge Isotope Laboratories | | Isotopic labeling |
| Other | Deuterium Oxide (99.8%) | Sigma Aldrich | | Isotopic labeling |
| Other | 13C6 Glucose | Cambridge Isotope Laboratories | | Isotopic labeling |

*Continued on next page*

*Continued*

| Reagent type (species) or resource | Designation | Source or reference | Identifiers | Additional information |
|---|---|---|---|---|
| Other | D-Glucose-13C6,1,2,3,4,5,6,6-d7 | Cambridge Isotope Laboratories | | Isotopic labeling |
| Other | Quantifoil R2/2 | Ted Pella | | Cryo-EM support film |
| Software, algorithm | Leginon | PMID: 11121305 | RRID:SCR_016731 | Cryo-EM data collection |
| Software, algorithm | SerialEM | PMID: 16182563 | | Cryo-EM data collection |
| Software, algorithm | auto3dem | PMID: 17029842 | | Cryo-EM 3D reconstruction |
| Software, algorithm | RobEM | Timothy S Baker's lab, UCSD | | Cryo-EM 3D reconstruction |
| Software, algorithm | Phenix | PMID: 29872004 | RRID:SCR_014224 | Cryo-EM map fitting and analysis |
| Software, algorithm | UCSF Chimera | PMID: 15264254 | RRID:SCR_004097 | cryo-EM image visualization |
| Software, algorithm | NAMD | PMID: 16222654 | RRID:SCR_014894 | Homology modeling |
| Software, algorithm | CHARMM | PMID: 19444816 | RRID:SCR_014892 | Homology modeling |
| Software, algorithm | MMTSB Tool Set | PMID: 15099834 | | Homology modeling |
| Software, algorithm | ccpNmr analysis | PMID: 15815974 | RRID:SCR_016983 | NMR data analysis |
| Software, algorithm | Aria | PMID: 17121777 | | NMR data analysis |
| Software, algorithm | CS-ROSETTA | PMID: 19034676 | RRID:SCR_015701 | NMR data analysis |
| Software, algorithm | TALOS-N | PMID: 23728592 | | NMR data analysis |
| Software, algorithm | Tensor-2 | PMID: 10718609 | | NMR data analysis |
| Software, algorithm | Dali | PMID: 20457744 | RRID:SCR_013433 | NMR data analysis |
| Software, algorithm | NMRbox | PMID: 28445744 | RRID:SCR_014827 | NMR data analysis |

## Strains and media

Phage L, its bacterial host, and purification procedures to produce high-titer stocks were previously described (*Gilcrease et al., 2005*; *Tang et al., 2006*). Stocks were stored in a 10 mM Tris (pH 7.6) and 10 mM MgCl$_2$ buffer. LB Miller broth and LB agar (Invitrogen) were used for all experiments.

## Purification of P22 phage with various coat proteins

*Salmonella enterica* serovar Typhimurium strain DB7136 (leuA414am, hisC525am, su$^0$), expressing P22 coat protein from mutant pMS11 plasmids, were infected with P22 phage carrying amber mutations in gene five to stop production of phage-encoded coat protein as described previously (*Suhanovsky and Teschke, 2013*). This P22 strain also carried the c1-7 allele to prevent lysogeny. The resulting phages containing amino acid substitutions in coat protein were purified using standard protocols (*Gilcrease et al., 2005*; *Parent et al., 2012b*).

## Site-directed mutagenesis of P22 coat protein and phage L Dec protein

The plasmid pMS11 was mutated using Quikchange protocols as described previously (*D'Lima and Teschke, 2015*) to generate plasmids that express coat protein containing the following substitutions: E81R, P82S, R299E, P322S, E323R, E81R:E323R. All plasmids were Amp$^R$ (100 μg/mL) and IPTG inducible (1 mM). Sequences were confirmed at the RSTF Genomics Core at Michigan State University. Plasmid pDec was mutated using Quikchange protocols, and was used to express wild type and variant Dec, each with a C-terminal histidine tag. Phage P22 purification (*Gilcrease et al., 2005*; *Parent et al., 2012b*) and binding assays were performed as described (*Tang et al., 2006*; *Parent et al., 2012b*). To measure binding, purified Dec was added to P22 phages made with WT or variant coat proteins. Free Dec was separated from Dec bound to P22 particles by cesium chloride density gradient sedimentation as previously described (*Gilcrease et al., 2005*; *Tang et al., 2006*). Bands containing P22 were TCA-precipitated, and analyzed by SDS-PAGE. Coomassie-

stained gel bands corresponding to P22 coat protein and Dec were quantitated and normalized to the ratio found in native P22 particles when bound with Dec. All binding experiments were repeated two to four times. Representative data are shown.

## Cryo-EM imaging

Aliquots (~5 µL) of phage L virions were vitrified and examined using published procedures (*Baker et al., 1999*). Briefly, this involved applying samples to Quantifoil R2/2 holey grids that had been plasma cleaned for ~20 s in a Fischione model 1020 plasma cleaner. Grids were then blotted with Whatman filter paper for ~5 s, and plunged into liquid ethane. Samples were pre-screened for concentraton and purification quality in a JEOL JEM-2200FS TEM using a Gatan 914 specimen holder, using standard low-dose conditions as controlled by SerialEM (*Mastronarde, 2005*). High-resoluiton imaging on an FEI Titan Krios was performed at Florida State University. Micrographs were recorded on a Direct Electron DE-20 camera wth a capture rate of 25 frames per second using a total of 53 frames, and a final dose of ~27 e/$Å^2$ at a final pixel size of 1.26 Å. Movie correction was performed using the Direct Electron software package, v2.7.1 (*Wang et al., 2014*) on entire frames.

## 3D image reconstructions of icosahedral particles

Micrographs exhibiting minimal astigmatism were selected for further processing. The objective lens defocus settings used to record each data set ranged 0.35 to 2.49 µm. In total, the final reconstruction used 7879 of the best particles from 494 images. The program RobEM (http://cryoEM.ucsd.edu/programs.shtm) was used to estimate micrograph defocus and astigmatism, extract individual phage L particles, and to preprocess the images. 150 particle images were used as input to the random-model computation procedure to generate an initial 3D density map at ~25 Å resolution (*Yan et al., 2007a*). Each map was then used to initiate determination and refinement of particle orientations and origins for the complete set of images using the current version of AUTO3DEM (v4.01.07) (*Yan et al., 2007b*). Phases of the particle structure factor data were corrected to compensate for the effects caused by the microscope contrast-transfer function, as described (*Bowman et al., 2002*). The Fourier Shell Correlation (FSC$_{0.5}$) 'gold standard' criterion was used to estimate the resolution of the final 3D reconstructions (*Pintilie et al., 2016*). The global resolution of the entire virion density map was determined to be 4.2 Å, without applying a mask. Local resolution was estimated as previously described (*Cardone et al., 2013*). Map segmentation was performed using the Segger tool in Chimera (*Goddard et al., 2007*). Phenix was used for autosharpening, atomic model fitting, and map validation for the segmented coat and Dec densities (*Afonine et al., 2018*). A portion of the native phage L capsid map (segmented coat and Dec density) has been deposited in the EMDB database (accession number EMD-9392). Graphical representations were generated using the UCSF Chimera visualization software package (*Goddard et al., 2007*).

## NMR characterization of the structure and dynamics of monomeric Dec

Samples of recombinant Dec enriched in $^{15}N$, $^{13}C$ and $^2H$ isotopes for NMR studies were expressed in *E. coli* and purified as described (*Newcomer et al., 2018*). To obtain samples suitable for NMR, 0.3 to 0.5 mM Dec was unfolded to pH 2 for 20 min, followed by refolding to pH 4.0 in 20 mM sodium acetate buffer containing 50 mM NaCl and 1 mM EDTA. The acid-unfolding/refolding procedure converted Dec to a monomer as monitored by native gel electrophoresis, size exclusion chromatography and $^{15}N$ relaxation data (*Newcomer et al., 2018*). All NMR data were collected for samples held at a temperature of 33°C. Virtually complete NMR assignments for monomeric Dec (>98% of backbone resonances) were obtained using a suite of 3D NMR experiments (*Newcomer et al., 2018*) and have been deposited in the Biological Magnetic Resonance Bank (http://www.bmrb.wisc.edu/) with the accession number 27435.

NMR structures for Dec were calculated with the program ARIA v. 2.3.1 (*Bardiaux et al., 2012*) using the experimental restraints summarized in *Table 2*. Backbone (φ,ψ) and side chain (χ1) torsion angles were calculated from assigned HN, N, Hα, Cα, Cβ, and CO NMR chemical shifts using the program TALOS-N (*Shen et al., 2009*). NOE-based distance restraints were obtained from 3D $^{15}N$- and $^{13}C$-NOESY experiments collected on $^{15}N/^{13}C$-labeled Dec samples, with or without 50% fractional deuteration. The NOESY experiments were collected on an 800 MHz spectrometer equipped with a cryogenic probe, using a mixing time of 100 ms. Hydrogen bond restraints were included

based on H-bond donors and acceptors identified in a long-range HNCO experiment (*Cordier and Grzesiek, 1999*) recorded on a $^2$H,$^{13}$C,$^{15}$N triple-labeled Dec sample in TROSY mode at 800 MHz, and H-bond donors inferred from amide proton protection in $^1$H to $^2$H hydrogen exchange experiments. Water refinement in the program ARIA (*Bardiaux et al., 2012*) was used as a final optimization step for the NMR structures.

Backbone dynamics of Dec were characterized using $^{15}$N NMR R1, R2, and $^1$H-$^{15}$N NOE relaxation data recorded at 800 MHz. R1 rates were obtained using interleaved relaxation delays of 0.05, 0.13, 0.21, 0.49, 0.57, 0.71 and 0.99 s. R2 rates were determined using interleaved relaxation delays of 0.01, 0.03, 0.05, 0.07, 0.09, 0.11, and 0.15 s. A 2 s pre-acquisition delay was used for recovery to thermal equilibrium. $^1$H-$^{15}$N NOE values were determined from the ratio of crosspeak intensities in a spectrum for which the proton signals were saturated for 2.5 s and a control spectrum in which the saturation period was replaced by a pre-acquisition delay of equivalent length. The processing and analysis of relaxation parameters was done according to published protocols (*Alexandrescu and Shortle, 1994*). Model-free analyses (*Lipari and Szabo, 1982*) of the $^{15}$N relaxation data were performed with the program Tensor2 (*Dosset et al., 2000*), yielding an optimal global isotropic rotational correlation time of 7.4 ns.

## Flexible fitting of coat protein and Dec into the cryo-EM map

Initial structures of the capsid proteins were built via homology modeling based on the available structure for P22 (PDB ID 5UU5; *Hryc et al., 2017*). Using the symmetry operations in the PDB structure, 1/8$^{th}$ of the capsid was constructed to cover the entire cryo-EM density map for phage L. To obtain an initial structure, the entire capsid complex was fitted via rigid-body docking to the density map using UCSF Chimera (*Goddard et al., 2007*).

After docking and refining the NMR-derived N-terminal OB-fold of Dec, a model for the helix spanning residues 78 to 86, and a linker was added to cover residues 87–92 before connecting with a homology model for the C-terminus based on a fragment from the bacteriophage T4 proximal long tail fiber protein gp34 (PDB ID: 4UXE; *Granell et al., 2017*). The EM density lacks sufficient resolution for reliably modeling the C-terminal part of Dec in terms of side chain orientations but indicates a barrel-shaped structure consistent with a β-helix seen in many bacteriophage tail structures. The modeled β-helix is also consistent with the β-sheet secondary structure predicted for the C-terminus of Dec from its sequence (*Tang et al., 2006*; *Parent et al., 2012b*; *Newcomer et al., 2018*). The selection of this structure as a template was based on the dimension of the β-helix that was narrower than most β-helices found in other tail structures, and in better agreement with the EM density. The complete model of the Dec trimer was then optimized against the EM density via flexible fitting using the MDFF protocol within NAMD (*Trabuco et al., 2008*). During the flexible fit, residues 1–77 were restrained at Cα and Cβ positions and the helices in each subunit spanning from 79 to 86 as well as the trimeric β-helix involving residues above 104 were internally restrained Cα positions. This allowed the C-terminal part of the trimer to move separately from the N-terminus to find the best fit to the EM density while preserving the secondary structure motifs. We note, that while the model for the N-terminal part of Dec is supported by NMR data and a good fit to the relatively high-resolution EM density, the model for the C-terminus is speculative due to a lack of high-resolution experimental data, therefore we only show the Cα backbone.

## Acknowledgements

We thank Timothy S Baker (University of California, San Diego), Gabriel C Lander and John E Johnson (The Scripps Research Institute) for advice and support during the preliminary phase of this project, Giovanni Cardone for support in the local resolution analysis, Prof. Angela Gronenborn (U Pittsburgh School of Medicine) for useful discussion, and Anne R Kaplan for help with NMR structure calculations of Dec. This material is based upon work supported by the AAAS Marion Milligan Mason Award for Women in the Chemical Sciences to KNP, by grants NIH GM084953 and GM126948 to MF, NIH grant R01 GM076661 and a grant from the UConn Research Excellence Program to ATA and CMT. High-resolution cryo-electron microscopy data were collected at Florida State University, and the FSU facility is supported by the following grants: S10 OD018142 and S10 RR025080 under PI Ken Taylor. We would like to thank Kaillathe 'Pappan' Padmanabhan for his assistance in the setup and maintenance of our computational resources. We thank Dr. Sundharraman Subramanian for help

with COOT. Finally, we gratefully acknowledge the NVIDIA Corporation with the donation of the Titan V GPU used for this research.

## Additional information

### Funding

| Funder | Grant reference number | Author |
|---|---|---|
| National Institute of General Medical Sciences | GM084953 | Michael Feig |
| National Institute of General Medical Sciences | GM126948 | Michael Feig |
| National Institute of General Medical Sciences | R01 GM076661 | Carolyn M Teschke Andrei T Alexandrescu |
| University of Connecticut | UConn Research Excellence Program | Carolyn M Teschke Andrei T Alexandrescu |
| American Association for the Advancement of Science | Marion Milligan Mason Award for Women in the Chemical Sciences | Kristin N Parent |
| National Institute of General Medical Sciences | GM110185 | Kristin N Parent |

The funders had no role in study design, data collection and interpretation, or the decision to submit the work for publication.

### Author contributions

Rebecca L Newcomer, Formal analysis, Investigation, Methodology, Writing—review and editing; Jason R Schrad, Conceptualization, Data curation, Formal analysis, Investigation, Methodology, Writing—review and editing; Eddie B Gilcrease, Resources, Investigation, Methodology; Sherwood R Casjens, Conceptualization, Resources, Formal analysis, Supervision, Funding acquisition, Investigation, Writing—review and editing; Michael Feig, Conceptualization, Resources, Software, Formal analysis, Investigation, Methodology, Writing—review and editing; Carolyn M Teschke, Resources, Supervision, Funding acquisition, Project administration, Writing—review and editing; Andrei T Alexandrescu, Conceptualization, Data curation, Software, Formal analysis, Supervision, Funding acquisition, Validation, Investigation, Methodology, Project administration, Writing—review and editing; Kristin N Parent, Conceptualization, Resources, Data curation, Software, Formal analysis, Supervision, Funding acquisition, Validation, Investigation, Visualization, Methodology, Writing—original draft, Project administration, Writing—review and editing

### Author ORCIDs

Rebecca L Newcomer (iD) https://orcid.org/0000-0002-6630-4926
Michael Feig (iD) http://orcid.org/0000-0001-9380-6422
Carolyn M Teschke (iD) http://orcid.org/0000-0001-6420-4895
Andrei T Alexandrescu (iD) https://orcid.org/0000-0002-8425-9276
Kristin N Parent (iD) http://orcid.org/0000-0002-6095-0628

### Decision letter and Author response

Decision letter https://doi.org/10.7554/eLife.45345.024
Author response https://doi.org/10.7554/eLife.45345.025

## Additional files

### Supplementary files

• Transparent reporting form
DOI: https://doi.org/10.7554/eLife.45345.018

## Data availability

cryo-EM data has been deposited to the EMDB under accession number EMD-9392 NMR data has been deposited to the PDB under accession number 6E3C.

The following datasets were generated:

| Author(s) | Year | Dataset title | Dataset URL | Database and Identifier |
|---|---|---|---|---|
| Newcomer RL, Schrad JR, Gil-crease EB, Casjens SR, Feig M, Teschke CM, Alex-andrescu AT, Par-ent KN | 2019 | NMR data | http://www.rcsb.org/ structure/6E3C | Protein Data Bank, 6E3C |
| Newcomer RL, Schrad JR, Gil-crease EB, Casjens SR, Feig M, Teschke CM, Alex-andrescu AT, Par-ent KN | 2019 | Cryo-EM data | http://www.ebi.ac.uk/ pdbe/entry/emdb/EMD-9392 | Electron Microscopy Data Bank, EMD-9392 |

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
