## [Decision Letter]

[Editors’ note: a previous version of this study was rejected after peer review, but the authors submitted for reconsideration. The first decision letter after peer review is shown below.]

Thank you for submitting your work entitled "How phage L capsid decoration protein distinguishes between nearly identical binding sites on an icosahedral virion" for consideration by *eLife*. Your article has been reviewed by three peer reviewers, one of whom is a member of our Board of Reviewing Editors, and the evaluation has been overseen by a Senior Editor. The reviewers have opted to remain anonymous.

Our decision has been reached after consultation between the reviewers. Based on these discussions and the individual reviews below, we regret to inform you that your work will not be considered further for publication in *eLife*.

All reviewers felt that it is of interest to understand discrimination of QE locations in capsids. One reason might be to understand immune recognition. Another might be to be able to direct assembly toward different T numbers. Unfortunately, the reviewers were not convinced that the paper gave insights into the structural basis for this in phage L, and whether this might be generalized to other viruses. Given that *eLife* does not make major revision decisions where substantial new work would be required, the only option was to reject the current submission. However, this does not preclude the future resubmission of a revised manuscript that fully addresses the concerns raised in this review.

*Reviewer #1:*

This manuscript presents structural studies of phage L capsid and capsid decoration proteins, using NMR, Cryo-EM and other biochemical approaches. Icosahedral virus capsid shells often have decoration proteins that help stabilize the virion structure against environmental insults. Among dsDNA tailed phages, those decoration proteins are known to have two common protein folds: one is the Ig domain fold such as pb10 in phage T5, and the other is a trimer with a β-tulip and α/β subdomain such as gpD in lambda phage. Now, in this manuscript, Newcomer et al. report a capsid protein Dec in Phage L that has an OB-fold, which is now the third fold of the dsDNA virus decoration proteins. In this paper, the Dec protein structure was obtained from NMR combined with homology modeling and then docked into the cryo-EM map. Interestingly, from the cryo-EM reconstruction the authors showed that the Dec proteins only selectively bind to the "quasi-three-fold" between hexons, but not to the "true three-fold" or the "quasi-three-fold" between hexons and a penton. The authors also try to explain the mechanism of Dec binding specificity through some mutagenesis studies and MD-based structural analysis. Overall the NMR part appears to be well designed and executed. On the other hand, the cryo-EM part was poorly presented, and there are several major issues that must be addressed.

1) The stated resolution of the cryo-EM map is ambiguous and misleading. The authors report a 4.2 Å overall resolution of the icosahedral reconstruction (I assume this means no mask was applied for this estimation, and the authors need to clarify this in the paper). But from Figure 1C, one can clearly tell the map resolution for the Dec protein part is probably around 8~15 Å, much worse than 4.2 Å. Also, the half-map FSC is measuring self-agreement between two maps rather than actual resolution. Therefore the authors need to measure the Model:Map FSC for the capsid proteins and Dec separately, and include those numbers in the paper.

2) The features in this cryo-EM map do not look like a typical 4.2 Å map. For example, one usually sees side chain densities on a helix and β-strand separation at this resolution. But these were not seen in the map provided by the authors. Therefore, it is either the authors over-estimated the cryo-EM resolution, or the map has not been properly sharpened. With the map quality as seen, it is hard to believe the detailed amino acid site analysis presented in Figure 5. This also suggests that some of the mutagenesis results can simply be explained by an inaccurate atomic model.

3) Both the capsid protein and Dec models sent from the authors have poor protein geometries, meaning they were poorly refined. Both models even have cis-peptides in the structure, which is unacceptable. The authors need to put a table in the main text listing protein geometries (bonds RMSD, Ramachandran statistics, etc.) of the deposited cryo-EM models and maps, and those statistics need to reach the common standard of the field.

4) One of the most interesting points in the paper involves the Dec selectivity on the Phage L capsid. But why were the mutagenesis experiments done using Phage P22, which doesn't have Dec?

5) The structural basis for the selective binding of Dec to capsids was poorly discussed. The MD-based simulations are not helpful in elucidating the mechanism. First, the authors need to understand that the hexons in the capsid do not necessarily have C6 symmetry. This is true from the model sent by the authors, and this means the "quasi-three-fold" and "true three-fold" may have different surface interfaces to Dec, which is the simplest explanation for Dec selectivity. With a 4.2 Å map, the author should be able to compare the surface using real cryo-EM map/models, rather than some MD simulations.

*Reviewer #2:*

The manuscript "How phage L capsid decoration protein distinguishes between nearly identical binding sites on an icosahedral virion" by Newcomer et al. looks at the binding of Dec on the capsid exterior surface in its host phage L, which varies according to position on the capsid surface. In addition, Dec binds to phage P22 due to 99.6% identity in coat protein sequence with that of phage L, although P22 doesn't encode its own Dec homolog. Capsid structures determined by high-resolution cryo-electron microscopy are combined with an NMR solution structure of monomeric Dec, MD simulations, and mutations across the P22 core and Dec interface to reveal the selectivity of Dec binding. Although this is a rather specific example from one of many similar phages, it nonetheless reflects a more general question of the fidelity of protein-protein binding that can also be found in other cellular contexts and for regulating function. Thus, it's a useful illustration of a general process that is readily studied in the tractable context of high-symmetry phage particles that are easily expressed and purified. However, while identifying several residues across multiple copies of the coat protein and in Dec that are important for binding, as well as the structural key to the binding selectivity, its not entirely clear what general principles have been discovered. Why does phage L have a Dec protein, but P22 has none? Why is Dec binding selective at all – ie, what difference is there between Dec binding nowhere, everywhere, and in the specific location identified? Nonetheless, as a model system the structural and binding results make an interesting story that is well supported by the data and offers a foundation for continuing to pursue important questions.

The manuscript could be greatly improved by correcting numerous errors, better wording, and a more concise presentation – including the Introduction, and NMR section in Results that descends into Discussion. Some examples are listed below, but others abound. A thorough read by all authors might really tighten this up.

1) Be careful to specify the scope of statement – e.g., the Abstract starts "The major coat proteins of most dsDNA tailed phages and viruses form capsids by a mechanism that includes active packaging of the dsDNA genome into a precursor procapsid, and subsequent expansion/stabilization of the capsid." By "most" the implication is that some dsDNA tailed phages and viruses do *not* package the genome into a preformed capsid etc. And yet there is a well-defined group in which this is, as far as I know, true – dsDNA tailed phages and herpesviruses. Why not say that? (The second sentence about atmospheres of pressure is pretty meaningless – why not talk more relevantly of concentration?) The third sentence then says "Phages and viruses have evolved diverse strategies to strengthen their capsids, such as non-covalent binding of auxiliary "decoration" proteins." This appears to be true of the dsDNA tailed phages and herpesviruses, but not necessarily for capsids that must disassemble to deliver their genomes, such as the dsDNA polyoma/papillomaviruses (but may be true for adenoviruses, on the other hand). The point being to word these statements precisely so as to avoid inadvertently generalizing in ways that are simply not correct.

2) Another example of a poorly chosen term "…such that some coat proteins are in hexameric conformations and some are in pentameric conformations" where strictly they are in hexavalent locations (usually but not always organized as hexamers – adenovirus trimers and polyomavirus pentamers, for example – or in fact more tightly organized as dimers as in hepatitis B virus) and pentavalent location, although those are organized as pentamers. As above, if you properly specify the subset of "Viral icosahedral capsids" as pentamer-hexamer dsDNA tailed phages and herpesviruses, then your statements following aren't wrong.

Similarly, in the first paragraph of the Introduction, environmental stress (plus that from internal dsDNA).

"At least 10^31^ viruses are found in the biosphere" neglects the indirect argument involved – 10^30^ bacteria were estimated from water samples, with 10 phage for each bacterium leading to the 10^31^ figure. Since phage are expected to dominate eukaryotic viruses numerically, the statement should read "10^31^ viruses are estimated in the biosphere". And the 2005 citation to support this [Bamford, Grimes and Stuart, 2005] is well-preceded by Hendrix's 1999 "All the world's a phage” paper in PNAS – see the start of the Discussion.

3) The tight packaging of DNA "…facilitates delivery of the majority of the viral genome into the host" is supported by what evidence? The dsDNA is not a fluid released like champagne from an uncorked bottle, and in some cases only a small portion is released initially with the rest drawn out by transcription – e.g., phage T5.

4) Introduction, third paragraph: The reference to chainmail should not be the 1989 Popa et al. paper, which merely indicated that the HK97 major capsid protein was involved in "wholesale covalent crosslinking", but instead the Duda 1998 Cell paper that proposed the chainmail lattice, and/or the Wikoff et al. crystal structure paper of 2000 in Science that verified it with the atomically-modeled details.

5) "gpD stabilizes its capsid by binding at all three-fold symmetry axes [Lander et al., 2008; Yang, Maluf and Catalano, 2008]" – first, it binds at the local and icosahedral 3-fold symmetry positions (axes invokes icosahedral only), and note that the use of "true three-fold axis" in Figure 6 and elsewhere would be better described as "icosahedral 3-fold axis". Secondly the citations are to recent structural papers that do not demonstrate that binding gpD stabilizes the capsid – that would be much earlier, Sternberg and Weisberg, 1977 or Imber et al., 1980, both in JMB. Structural data were also available much earlier, from the first lambda capsid map by Dokland and Murialdo, 1995, to the gpD crystal structure and improved capsid map by Yang et al., 2000.

6) "P-loops" are not defined here, or anywhere. Indicate such domains on Figure 1—figure supplement 2.

7) "The twelfth penton vertex is occupied by the portal channel…" – first, its not really a "penton vertex" since no penton occupies it; and second, while it is indeed occupied by a channel it would be more correct would be to say that its occupied by the portal complex that connects to the tail apparatus.

8) The amino acid differences between phage L and P22 coat proteins are marked on Supplementary Figure 2D, but it would be useful to see them more in context by marking them (as spheres?) on Supplementary Figure 2E.

9) Supplementary Figure 4 colors one of the three subunits in rainbow according to the caption – this does not appear to the case as it is cyan and purple with some abrupt point of transition. Marking the N- and C-termini, and including the fold (one subunit) may help with this figure. The main text (subsection “Structures of the phageL coat lattice and capsid-bound Dectrimers”, third paragraph) suggests that cyan is the N-terminal region (OB-fold) and the purple is the C-terminal region (3-stranded β-helix and protruding central stalk). Please clarify in the legend.

10) Subsection “NMR structure of the Dec monomer has an OB-fold domain and an unfolded C terminus”, first and last paragraphs and legend to Supplementary Figure 4: Supplementary figure 5 should be Supplementary Figure 6. However, the two figures should probably be swapped as the NMR data (Supplementary Figure 6) are referred to before the Rosetta modeling (Supplementary Figure 5).

11) Confusing as written "To further characterize the dynamics of Dec, we collected NMR 15N-relaxation data (Supplementary Figure 5) and analyzed the dynamics of the monomeric protein in terms of the "Model-Free" formalism." Since the NMR data have already been described for the previous two pages, introducing the data collection again is weird. Perhaps it should read "To further characterize the dynamics of the monomeric Dec protein, we analyzed the NMR 15N-relaxation data (Supplementary Figure 5) in terms of the "Model-Free" formalism."

12) "We noted that coat proteins occupying different local conformations 290 (as a result of the quasi-equivalent capsid lattice) contribute different residues to the binding interface (Figure 5A, Video 3)." The difference in residues contributed is not apparent in either Figure 5A nor in the video.

13) Results, subsection “A structural basis for the discminatory binding of Dec to capsids”, does not include the pentamer-hexamer quasi-3-fold location, and why Dec does not bind there.

14) Subsection “Decoration proteins employ a variety of binding schemes to adhere to phage capsids”, might include a reference to the herpesvirus triplex binding.

15) "occupy similar spatial positions as the HK97 catalytic residues that crosslink to form the lattice "chainmail"." These residues would be better called the crosslinking residues as a catalyst by definition is not "used up" by the reaction – a third residue is proposed to act as a catalyst for the crosslinking reaction.

16) "the inherent asymmetry in the Dec trimer" – this Dec asymmetry is observed in the cryoEM data where the Dec trimer is bound to the capsid at positions that depart slightly from 3-fold symmetry, and presumably the homo-trimer is flexible enough to accommodate the capsid. It is not observed in the NMR data since that is of monomeric Dec. What then is the basis for saying that this asymmetry is inherent in the Dec trimer when is it rather more certainly inherent in the capsid at the positions Dec binds to? The strongest evidence might be that Dec doesn't bind at the icosahedral (strict) 3-fold site, but then it also doesn't bind at the local 3-fold adjacent to the pentamers (and this isn't explained – see #13 above).

17) The authors might consider mentioning earlier in the Introduction that gpD and Dec bind the capsids as homotrimers.

18) Some figures are not easily understood, including Figure 1 where the capsomers and the T=7 organization are hard to see (even Supplementary Figure 2 isn't entirely clear, but it is better).

*Reviewer #3:*

In this manuscript Newcomer et al. bring three approaches to bear on the question of how phage L Dec protein discriminates between true and quasi three-fold symmetry sites. The structure of phage L coat lattice and capsid bound Dec trimers is obtained by cryo-EM, an NMR structure of a partially refolded Dec monomer is obtained, and site directed mutagenesis is performed on P22 coat protein to assess the effect on Dec binding.

The techniques brought to bear are all high-resolution techniques and the cryo-EM and NMR experiments seem to be technically solid. In this reviewer's opinion the NMR studies of a partially refolded Dec monomer are something of a distraction from the focus of the manuscript.

Curiously, the mutagenesis experiments were performed using bacteriophage P22 coat protein rather than phage L. This reviewer appreciates the similarity between the two phage, and also the availability of systems for genetic manipulation and production of P22 phage. While it seems likely (based on the 2006 publication by Tang et al.) that binding is the same, constructing a P22 coat protein mutant containing the 4 phage L sequence alterations would seem to be straightforward and more compelling.

The major reservation this reviewer has is despite the effort and high-resolution structures obtained the end result is that discriminatory binding is "largely due to the surface area of contact with the capsid protein", a vague statement which could have been made even prior to this study.

There are no doubt valuable and intriguing findings in this manuscript. However, perhaps reflective of the interests of the multiple groups involved the focus drifts from discriminatory binding, to the novelty of the structure of the Dec protein itself, to unanswered questions about why some phage have decoration/stabilization proteins and some don't.

[Editors’ note: what now follows is the decision letter after the authors submitted for further consideration.]

Thank you for submitting your article "The phage L capsid decoration protein has a novel OB-fold and an unusual capsid binding strategy" for consideration by *eLife*. Your article has been reviewed favorably by three peer reviewers, one of whom is a member of our Board of Reviewing Editors, and the evaluation has been overseen by John Kuriyan as the Senior Editor. The reviewers have opted to remain anonymous.

The reviewers have discussed the reviews with one another and the Reviewing Editor has drafted this decision to help you prepare a revised submission.

Summary:

This is a revised version of the manuscript by Newcomer et al. in which they describe the structure of phage L, a P22-like dsDNA phage that, unlike P22, is stabilized by decoration proteins that bind to quasi-threefold, but not icosahedral threefold positions in the capsid. Furthermore, they determine the NMR solution structure of Dec, which has an OB fold, different from e.g. the gpD decoration protein of phage lambda. Additionally, mutational analysis was used to identify the residues involved in the CP-Dec binding interaction.

The authors have generally responded well to the previous reviewers' comments. Most importantly, they provide an improved map and more well refined model.

Essential revisions:

1) Discussion: "…decoration proteins for which atomic-resolution structures are known…" This fails to consider the Psu decoration protein of phage P4, which, like Dec, serves to stabilize the capsids and can be added to capsids extraneously, but, unlike Dec, binds across the center of hexamers (Isaksen et al., 1993, Dokland et al., 1993). The crystal structure of Psu has been determined (Banerjee et al., 2012) and has an α-helical fold, completely different from those of T4, T5, L or lambda. Presumably Psu belongs to an even more distinct lineage of decoration proteins.

2) Figure 1C is somewhat misleading, in that the color scheme suggests that the six subunits are organized around an icosahedral threefold axis. It would be more useful to color the subunits according to the quasi-symmetric location (A-G) and provide a key to understand which is which (e.g. by indicating on Figure 1A). Positions of symmetry axes (twofold?) should also be indicated.

3) Figure 6: I get that the two quasi-threefold and the icosahedral threefold position differ in the angle between subunits and that this probably determines whether Dec binds or not. However, the figure is rather messy and it is difficult to see where Dec would bind to the CP. Is there a better way to represent this so that the CP-Dec interaction can be seen in the side view? Maybe a thinner slice through the structure, depth cueing or something like that?

4) Furthermore, the table in Figure 6, which lists differences in distance between residue E81, does not convey the point made in the text (subsection “Decoration proteins employ a variety of binding schemes to adhere to phage capsids”, last paragraph) that the icosahedral threefold is the flattest, the Dec binding quasi-threefold is intermediate, and the pentamer-proximal position is the most angled. One might have presumed that the most angled positions that represent a "weak point in many capsids" would be the one where Dec binds. There does not seem to be much discussion of this interesting point, which may point to Dec's function – perhaps not to stabilize capsids after all?

5) Also, in Figure 6, the arc with the denotation 54.2Å is confusing. What is this the distance between? It seems to indicate an angle rather than a distance. Was it meant to be 54.2°? It might be helpful to list these angles differences between all three types of trimers.

6) I find it a really missed opportunity to demonstrate the claimed quality of the maps by not showing some region with excellent side-chain density with the fold model fit into it – not in the manuscript, supplementary figures or videos.

7) This rebuttal statement "We have de-emphasized the notion of discriminatory binding in our title and throughout the text" does not seem to be accurate. The Abstract states: "The Dec protein from the P22-like phage L has a highly unusual binding strategy that precisely distinguishes between nearly identical three-fold and quasi-three-fold sites of the icosahedral capsid." and "Here, we investigate how Dec is able to distinguish subtly different capsid binding sites with high specificity." This aspect has rather more prominence than claimed.

---

## [Author Response]

[Editors’ note: the author responses to the first round of peer review follow.]

All reviewers felt that it is of interest to understand discrimination of QE locations in capsids. One reason might be to understand immune recognition. Another might be to be able to direct assembly toward different T numbers. Unfortunately, the reviewers were not convinced that the paper gave insights into the structural basis for this in phage L, and whether this might be generalized to other viruses. Given that eLife does not make major revision decisions where substantial new work would be required, the only option was to reject the current submission. However, this does not preclude the future resubmission of a revised manuscript that fully addresses the concerns raised in this review.

Thank you for the suggestions and for the opportunity to make the text more generalizable to a broad audience. We have addressed these points in the Introduction and Discussion sections to include how this is generalizable to other viruses, nanomedicine, general assembly and stability mechanisms and also of great utility in nanomaterials design. We also emphasized the point that an OB-fold is a novel domain for decoration proteins and this is a fundamentally important finding within this work. We have indicated that the decoration proteins discussed specifically in our manuscript bind only after maturation, so they cannot affect T number.

Reviewer #1:[…]1) The stated resolution of the cryo-EM map is ambiguous and misleading. The authors report a 4.2 Å overall resolution of the icosahedral reconstruction (I assume this means no mask was applied for this estimation, and the authors need to clarify this in the paper). But from Figure 1C, one can clearly tell the map resolution for the Dec protein part is probably around 8~15 Å, much worse than 4.2 Å. Also, the half-map FSC is measuring self-agreement between two maps rather than actual resolution. Therefore the authors need to measure the Model:Map FSC for the capsid proteins and Dec separately, and include those numbers in the paper.

We have run map validation procedures using Phenix and have listed this information in the Materials and Methods section. By applying the autosharpening routine, the display of the map is much improved, and the new figures accurately reflect the high resolution. Indeed, this capsid reconstruction has varying resolution at different radii. We have tried to further emphasize this point where we talk about the local resolution assessment (originally Supplementary Figure 3, now moved to Figure 1—figure supplement 1). Selected statistics from the validation report are now shown in Table 1.

2) The features in this cryo-EM map do not look like a typical 4.2 Å map. For example, one usually sees side chain densities on a helix and β-strand separation at this resolution. But these were not seen in the map provided by the authors. Therefore, it is either the authors over-estimated the cryo-EM resolution, or the map has not been properly sharpened. With the map quality as seen, it is hard to believe the detailed amino acid site analysis presented in Figure 5. This also suggests that some of the mutagenesis results can simply be explained by an inaccurate atomic model.

We appreciate the point made by the reviewer and we acknowledge we were in error in supplying the raw data for review. We have now supplied a sharpened map, where the highresolution quality is easier to visualize.

After using the autosharpening routine within Phenix, the display of the map is much improved, and the new figures accurately reflect the high resolution. The detailed side chain information that guided the mutagenesis experimentation is well supported by three independent points:

1) The phage L coat protein density is of sufficient quality to determine a complete model de novo(see above and local resolution refinement Figure 1—figure supplement 1).

2) Phage L coat protein has 99.6% identity to P22 coat protein. The majority of our phage L model is within 1-1.5 Å RMSD from the high-resolution models derived from P22 data (3.3 Å resolution in Hryc et al., 2017 PNAS – https://www.ncbi.nlm.nih.gov/pubmed/28270620).

3) Our NMR data is derived independently. The NMR data accurately describes the N-terminal portion of Dec protein (residues 1-77), and docks in very well with the cryo-EM derived models. While we agree that the C-terminal β-helix is much lower resolution in our cryo-EM map, only the N-terminal OB fold of the Dec protein as docked into the trimer is necessary for designing/interpreting site-directed mutagenesis studies.

Therefore, a hypothesis regarding detailed side chain information where the coat protein contacts Dec is a perfectly reasonable one to make and to test. The experimental evidence from the mutagenesis data aligns with our predictions and gives additional strength to the atomic models of coat and Dec. In fact, ours is one of very few papers that experimentally tests decoration protein binding to capsids via mutagenesis, and we believe this significantly adds to the current understanding within the field. We have made it a point to more clearly state the value of this experiment throughout the text.

3) Both the capsid protein and Dec models sent from the authors have poor protein geometries, meaning they were poorly refined. Both models even have cis-peptides in the structure, which is unacceptable. The authors need to put a table in the main text listing protein geometries (bonds RMSD, Ramachandran statistics, etc.) of the deposited cryo-EM models and maps, and those statistics need to reach the common standard of the field.

We have used Phenix to perform real-space refinement within the sharpened density and have improved models. Furthermore, we have deposited the validation reports along with our models to the EMDB. Selected statistics from the validation report are now shown in Table 1. The full validation reports for the capsid protein are available to anyone who wants access under the accession number EMD-9392. The NMR structure has been deposited to the PDB and validation reports on that structure are also publically available under the accession number 6E3C.

4) One of the most interesting points in the paper involves the Dec selectivity on the Phage L capsid. But why were the mutagenesis experiments done using Phage P22, which doesn't have Dec?

This is a result of two considerations:

1) Using P22 drastically increased efficiency and reduced the cost of the experiments. Since P22 is a well-established model system, we had access to clear plaque mutants, amber mutants, non-sense tRNA suppressors, cloned plasmids, a variety of host strains etc. We would need to establish a complete genetic system in phage L in order to do these experiments. Since previous work showed that P22 and phage L capsids are virtually identical (Gilcrease et al., 2005, Tang et al., 2006, and Parent et al., 2010), we decided there was no justification to spend the time and money on the creation of an entirely new Phage L genetic system.

2) P22 is a well-established model system that is used by many laboratories for molecular display and nano-materials design. Dec is often used in these experiments with P22. Therefore, it is very useful to have the mutant binding data using P22 particles. See a large body of work by Trevor Douglas’ group (>30 publications) as well as nanomaterials work from other groups.

5) The structural basis for the selective binding of Dec to capsids was poorly discussed. The MD-based simulations are not helpful in elucidating the mechanism. First, the authors need to understand that the hexons in the capsid do not necessarily have C6 symmetry. This is true from the model sent by the authors, and this means the "quasi-three-fold" and "true three-fold" may have different surface interfaces to Dec, which is the simplest explanation for Dec selectivity. With a 4.2 Å map, the author should be able to compare the surface using real cryo-EM map/models, rather than some MD simulations.

Unfortunately, MD was the only way to answer this question, as there was no significant cryo-EM density of Dec at the true icosahedral three-fold site. Therefore, the only way we could possible examine how Dec might interact with the surface was computationally. We couldn’t use “real” cryo-EM map/models without evidence of real density.

In light of this reviewer’s concern over the use of computational methodology, we have removed this result from the current work.

Reviewer #2:The manuscript "How phage L capsid decoration protein distinguishes between nearly identical binding sites on an icosahedral virion" by Newcomer et al. looks at the binding of Dec on the capsid exterior surface in its host phage L, which varies according to position on the capsid surface. In addition, Dec binds to phage P22 due to 99.6% identity in coat protein sequence with that of phage L, although P22 doesn't encode its own Dec homolog. Capsid structures determined by high-resolution cryo-electron microscopy are combined with an NMR solution structure of monomeric Dec, MD simulations, and mutations across the P22 core and Dec interface to reveal the selectivity of Dec binding. Although this is a rather specific example from one of many similar phages, it nonetheless reflects a more general question of the fidelity of protein-protein binding that can also be found in other cellular contexts and for regulating function. Thus, it's a useful illustration of a general process that is readily studied in the tractable context of high-symmetry phage particles that are easily expressed and purified. However, while identifying several residues across multiple copies of the coat protein and in Dec that are important for binding, as well as the structural key to the binding selectivity, its not entirely clear what general principles have been discovered. Why does phage L have a Dec protein, but P22 has none? Why is Dec binding selective at all – ie, what difference is there between Dec binding nowhere, everywhere, and in the specific location identified? Nonetheless, as a model system the structural and binding results make an interesting story that is well supported by the data and offers a foundation for continuing to pursue important questions.The manuscript could be greatly improved by correcting numerous errors, better wording, and a more concise presentation – including the Introduction, and NMR section in Results that descends into Discussion. Some examples are listed below, but others abound. A thorough read by all authors might really tighten this up.

Thank you for finding value in the story and the data. We have significantly improved the Introduction and Discussion and have removed extraneous detail in the NMR section. We hope this makes the manuscript easier to read. We also think it’s interesting that phage L encodes a Dec protein while phage P22 does not. Given that the phage P22 capsid is very stable and its coat protein has only 4 of 430 a.a. different from phage L, it is puzzling that the latter phage would make an extra stabilizing protein given the pressures on viruses to minimize their genomes for efficient packaging. One possible solution to this paradox is if Dec had a second function besides capsid stabilization such as cell recognition. We have evidence that Dec might bind to target cells, presumably through carbohydrate moieties emanating from the target cell surfaces. From a structural point of view, both the OB-fold part of Dec and the C-terminal β-helix could be carbohydrate binding motifs. Because our data at this point is preliminary and pertains to a different scope – capsid-binding of Dec, we did not include it in the present manuscript. If Dec indeed turns out to have a second targetcell binding function, the selectivity of Dec binding could serve a structural role – namely different surface patterning on the icosahedral lattice, that could be used for target-cell binding selectively.

1) Be careful to specify the scope of statement – eg, the Abstract starts "The major coat proteins of most dsDNA tailed phages and viruses form capsids by a mechanism that includes active packaging of the dsDNA genome into a precursor procapsid, and subsequent expansion/stabilization of the capsid." By "most" the implication is that some dsDNA tailed phages and viruses do not package the genome into a preformed capsid etc. And yet there is a well-defined group in which this is, as far as I know, true – dsDNA tailed phages and herpesviruses. Why not say that? (The second sentence about atmospheres of pressure is pretty meaningless – why not talk more relevantly of concentration?) The third sentence then says "Phages and viruses have evolved diverse strategies to strengthen their capsids, such as non-covalent binding of auxiliary "decoration" proteins." This appears to be true of the dsDNA tailed phages and herpesviruses, but not necessarily for capsids that must disassemble to deliver their genomes, such as the dsDNA polyoma/papillomaviruses (but may be true for adenoviruses, on the other hand). The point being to word these statements precisely so as to avoid inadvertently generalizing in ways that are simply not correct.

Thank you for the suggestion. We have addressed these points by specifically stating “dsDNA phages and herpesviruses” to be explicit.

2) Another example of a poorly chosen term "…such that some coat proteins are in hexameric conformations and some are in pentameric conformations" where strictly they are in hexavalent locations (usually but not always organized as hexamers! – adenovirus trimers and polyomavirus pentamers, for example – or in fact more tightly organized as dimers as in hepatitis B virus) and pentavalent location, although those are organized as pentamers. As above, if you properly specify the subset of "Viral icosahedral capsids" as pentamer-hexamer dsDNA tailed phages and herpesviruses, then your statements following aren't wrong.

We changed our text to be more explicit.

Similarly, in the first paragraph of the Introduction, environmental stress (plus that from internal dsDNA).

We changed our text to add the point regarding dsDNA packaging.

"At least 10^31^ viruses are found in the biosphere" neglects the indirect argument involved – 10^30^ bacteria were estimated from water samples, with 10 phage for each bacterium leading to the 10^31^ figure. Since phage are expected to dominate eukaryotic viruses numerically, the statement should read "10^31^ viruses are estimated in the biosphere". And the 2005 citation to support this [Bamford, Grimes and Stuart, 2005] is well-preceded by Hendrix's 1999 "All the world's a phage” paper in PNAS – see the start of the Discussion.

Unfortunately, we have to disagree with this reviewer on this point. The subject of the Hendrix 1999 paper is evolutionary relationships that align DNA sequences and protein sequences to infer relatedness among diverse phages. There are no data presented in that work that serves as an estimate for the total number of phages in the biosphere.

The Hendrix paper does however reference a 1989 paper from Bergh et al. that estimates the abundances of viruses in aquatic environments. In addition, there was an updated report in 2000 from Wommack et al. We have replaced the Bamford reference with these two papers to provide references that precede 2005.

3) The tight packaging of DNA "…facilitates delivery of the majority of the viral genome into the host." Is supported by what evidence? The dsDNA is not a fluid released like champagne from an uncorked bottle, and in some cases only a small portion is released initially with the rest drawn out by transcription – e.g., phage T5.

There is an enormous body of evidence to support the idea that the pressure inside the capsid is sufficient to eject some or in many cases most of the dsDNA chromosome for a variety of phages. We have added these references to the paper and clarified the statement that this phenomenon relates to “many phages and herpesviruses” so as not to imply that this is true for T5:

Some seminal original research papers on pressure and ejection of phages and Herpesviruses:

1) Evilevitch, 2018

2) Kellermeyer et al., 2018

3) Bauer et al., 2013

4) Bauer et al., 2015

5) Jin et al. 2015 Virology:

https://www.ncbi.nlm.nih.gov/pubmed/?term=jin+parent+gelbart

6) Gelbart and Knobler, 2009

7) Grayson et al., 2006 Virology: https://www.ncbi.nlm.nih.gov/pubmed/16469346

8) Kindt et al., 2001

Some review papers:

1) Molineux and Panja 2013 Nat Rev Microbiol:

https://www.ncbi.nlm.nih.gov/pubmed/23385786

2) Roos et al., 2007 Cell Mol Life Sci: https://www.ncbi.nlm.nih.gov/pubmed/17440680

4) Introduction, third paragraph: The reference to chainmail should not be the 1989 Popa et al. paper, which merely indicated that the HK97 major capsid protein was involved in "wholesale covalent crosslinking", but instead the Duda 1998 Cell paper that proposed the chainmail lattice, and/or the Wikoff et al. crystal structure paper of 2000 in Science that verified it with the atomically-modeled details.

Thank you for the suggestion. We have added the Duda and Wikoff references to the end of the sentence, but also kept the Popa reference where we state that the capsid is stabilized by crosslinks as this was the first paper to demonstrate covalent crosslinking phenomenon.

5) "gpD stabilizes its capsid by binding at all three-fold symmetry axes [Lander et al., 2008; Yang, Maluf and Catalano, 2008]" – first, it binds at the local and icosahedral 3-fold symmetry positions (axes invokes icosahedral only), and note that the use of "true three-fold axis" in Figure 6 and elsewhere would be better described as "icosahedral 3-fold axis". Secondly the citations are to recent structural papers that do not demonstrate that binding gpD stabilizes the capsid – that would be much earlier, Sternberg and Weisberg, 1977, or Imber et al., 1980, both in JMB. Structural data were also available much earlier, from the first lambda capsid map by Dokland and Murialdo, 1995, to the gpD crystal structure and improved capsid map by Yang et al., 2000.

Thank you for the suggestion. We have replaced all instances of “true three-fold” with “icosahedral three-fold”.

Thank you also for the suggestions of additional references. It was not our intention to leave out critical findings, and we have happily added the Sternberg and Weisberg, Imber, and Dokland Murialdo citations in the Introduction where we first describe gpD.

6) "P-loops" are not defined here, or anywhere. Indicate such domains on Figure 1—figure supplement 2.

Label added.

7) "The twelfth penton vertex is occupied by the portal channel…" – first, its not really a "penton vertex" since no penton occupies it, how about dropping "penton"; and second, while it is indeed occupied by a channel it would be more correct would be to say that its occupied by the portal complex that connects to the tail apparatus.

The word “penton” has been removed. “Channel” has been changed to “complex”.

8) The amino acid differences between phage L and P22 coat proteins are marked on Supplementary Figure 2D, but it would be useful to see them more in context by marking them (as spheres?) on Supplementary Figure 2E.

It proved very difficult to show this well in a 2D view, so instead we made a video showing the location of these differences surrounding a Dec binding site at a quasi three-fold axis (new Video 1).

9) Supplementary Figure 4 colors one of the three subunits in rainbow according to the caption – this does not appear to the case as it is cyan and purple with some abrupt point of transition. Marking the N- and C-termini, and including the fold (one subunit) may help with this figure. The main text (subsection “Structures of the phageL coat lattice and capsid-bound Dectrimers”, third paragraph) suggests that cyan is the N-terminal region (OB-fold) and the purple is the C-terminal region (3-stranded β-helix and protruding central stalk). Please clarify in the legend.

This point is no longer relevant as we have removed this figure from the manuscript.

10) Subsection “NMR structure of the Dec monomer has an OB-fold domain and an unfolded C terminus”, first and last paragraphs and legend to Supplementary Figure 4: Supplementary Figure 5 should be Supplementary Figure 6. However, the two figures should probably be swapped as the NMR data (Supplementary Figure 6) are referred to before the Rosetta modeling (Supplementary Figure 5).

In response to several comments by the reviewers, we have reordered the text and the corresponding figures to match in several places.

11) Confusing as written "To further characterize the dynamics of Dec, we collected NMR 15N-relaxation data (Supplementary Figure 5) and analyzed the dynamics of the monomeric protein in terms of the "Model-Free" formalism." Since the NMR data have already been described for the previous two pages, introducing the data collection again is weird. Perhaps it should read "To further characterize the dynamics of the monomeric Dec protein, we analyzed the NMR 15N-relaxation data (Supplementary Figure 5) in terms of the "Model-Free" formalism."

Thank you for the suggestion. Fixed.

12) "We noted that coat proteins occupying different local conformations 290 (as a result of the quasi-equivalent capsid lattice) contribute different residues to the binding interface (Figure 5A, Video 3)." The difference in residues contributed is not apparent in either Figure 5A nor in the video.

We have adjusted the figure, the videos and the legend to make this more clear.

13) Results, subsection “A structural basis for the discminatory binding of Dec to capsids”, does not include the pentamer-hexamer quasi-3-fold location, and why Dec does not bind there.

We have de-emphasized the notion of discriminatory binding in our title and throughout the text.

14) Subsection “Decoration proteins employ a variety of binding schemes to adhere to phage capsids”, might include a reference to the herpesvirus triplex binding.

We added two references: an original structure paper that first shows the triplexes (Zhou et al., 1994) and a review article (Heming et al., 2017).

15) "occupy similar spatial positions as the HK97 catalytic residues that crosslink to form the lattice "chainmail"." These residues would be better called the crosslinking residues as a catalyst by definition is not "used up" by the reaction – a third residue is proposed to act as a catalyst for the crosslinking reaction.

We have changed “catalytic” to “crosslinking”.

16) "the inherent asymmetry in the Dec trimer" – this Dec asymmetry is observed in the cryoEM data where the Dec trimer is bound to the capsid at positions that depart slightly from 3-fold symmetry, and presumably the homo-trimer is flexible enough to accommodate the capsid. It is not observed in the NMR data since that is of monomeric Dec. What then is the basis for saying that this asymmetry is inherent in the Dec trimer when is it rather more certainly inherent in the capsid at the positions Dec binds to? The strongest evidence might be that Dec doesn't bind at the icosahedral (strict) 3-fold site, but then it also doesn't bind at the local 3-fold adjacent to the pentamers (and this isn't explained – see #13 above).

We have removed this statement.

17) The authors might consider mentioning earlier in the Introduction that gpD and Dec bind the capsids as homotrimers.

We have added this detail to the Introduction.

18) Some figures are not easily understood, including Figure 1 where the capsomers and the T=7 organization are hard to see (even Supplementary Figure 2 isn't entirely clear, but its better).

We have replaced Figure 1A with a cartoon schematic that better shows T = 7 organization. Supplementary Figure 2 is now removed in the revised text.

Reviewer #3:In this manuscript Newcomer et al. bring three approaches to bear on the question of how phage L Dec protein discriminates between true and quasi three-fold symmetry sites. The structure of phage L coat lattice and capsid bound Dec trimers is obtained by cryo-EM, an NMR structure of a partially refolded Dec monomer is obtained, and site directed mutagenesis is performed on P22 coat protein to assess the effect on Dec binding.The techniques brought to bear are all high-resolution techniques and the cryo-EM and NMR experiments seem to be technically solid. In this reviewer's opinion the NMR studies of a partially refolded Dec monomer are something of a distraction from the focus of the manuscript.

We reworked the text surrounding the NMR data to highlight the important parts, but to decrease on the extraneous details. We have recent unpublished data that freshly purified Dec trimer slowly dissociates to monomers over a period of about a week in solution. Thus the Dec trimer seems to be a metastable kinetically trapped state in solution. Whether the capsids or other intracellular factors stabilize the Dec trimer is an open question, but our results suggest that the monomeric form of Dec is a mechanistically important species. We are currently investigating the mechanism of Dec trimer formation and dissociation. Lastly, the NMR data for the OB-fold was the basis for building the trimer into the cryo-EM density map as well as guiding site-directed mutagenesis, and thus us critical to the work.

Curiously, the mutagenesis experiments were performed using bacteriophage P22 coat protein rather than phage L. This reviewer appreciates the similarity between the two phage, and also the availability of systems for genetic manipulation and production of P22 phage. While it seems likely (based on the 2006 publication by Tang et al.) that binding is the same, constructing a P22 coat protein mutant containing the 4 phage L sequence alterations would seem to be straightforward and more compelling.

We chose not to make these substitutions as none of them are involved in Dec binding, or near the binding sites. If any were involved in this binding interface we would have expected the binding properties of Dec (e.g. the saturation, affinity, and the binding patterns) to be affected when comparing binding to P22 or L. Additionally, each of the amino acids that are changed in L are found in some P22 stocks as phenotypically silent mutations (Teschke lab, unpublished), suggesting that these minor differences play no role in the way the phage capsids assemble or function. We have added this description to the text.

The major reservation this reviewer has is despite the effort and high-resolution structures obtained the end result is that discriminatory binding is "largely due to the surface area of contact with the capsid protein", a vague statement which could have been made even prior to this study.There are no doubt valuable and intriguing findings in this manuscript. However, perhaps reflective of the interests of the multiple groups involved the focus drifts from discriminatory binding, to the novelty of the structure of the Dec protein itself, to unanswered questions about why some phage have decoration/stabilization proteins and some don't.

We have emphasized the novelty of the Dec OB-fold, as well as the discovery of which residues contribute to capsid binding. These observations would not have been possible without the high-resolution work contained here.

We have de-emphasized the discussion regarding selectivity of binding in some places, although we have added a new figure (new Figure 6) that shows how the coat protein structures change at the different sites to make this more obvious.

[Editors' note: the author responses to the re-review follow.]

Essential revisions:1) Discussion: "…decoration proteins for which atomic-resolution structures are known…" This fails to consider the Psu decoration protein of phage P4, which, like Dec, serves to stabilize the capsids and can be added to capsids extraneously, but, unlike Dec, binds across the center of hexamers (Isaksen et al., 1993, Dokland et al., 1993). The crystal structure of Psu has been determined (Banerjee et al., 2012) and has an α-helical fold, completely different from those of T4, T5, L or lambda. Presumably Psu belongs to an even more distinct lineage of decoration proteins.

Thank you for this addition. We have added these references to the Discussion (Isaksen, Dokland and Lindqvist, 1993; Dokland et al., 1993; Banerjee et al., 2012) and included a mention that Psu is from another category of decoration proteins.

2) Figure 1C is somewhat misleading, in that the color scheme suggests that the six subunits are organized around an icosahedral threefold axis. It would be more useful to color the subunits according to the quasi-symmetric location (A-G) and provide a key to understand which is which (e.g. by indicating on Figure 1A). Positions of symmetry axes (twofold?) should also be indicated.

We have adjusted the figure according to the color scheme recommended by the reviewers and added an enlarged area to show the fit of the map and the high resolution allowing us to view side chains. The figure legend has been edited accordingly.

3) Figure 6: I get that the two quasi-threefold and the icosahedral threefold position differ in the angle between subunits and that this probably determines whether Dec binds or not. However, the figure is rather messy and it is difficult to see where Dec would bind to the CP. Is there a better way to represent this so that the CP-Dec interaction can be seen in the side view? Maybe a thinner slice through the structure, depth cueing or something like that?

We have updated the figure to show additional views including an overlay, as well as two individual views of the quasi three-fold between hexamers (light blue) as well as the quasi three-fold site between hexamers and pentamers (dark blue). We also included the Dec density (lavender) in this new figure to emphasize the differences between these two quasi equivalent environments.

4) Furthermore, the table in Figure 6, which lists differences in distance between residue E81, does not convey the point made in the text (subsection “Decoration proteins employ a variety of binding schemes to adhere to phage capsids”, last paragraph) that the icosahedral threefold is the flattest, the Dec binding quasi-threefold is intermediate, and the pentamer-proximal position is the most angled.

We have expanded the Discussion in a few places to address the point regarding E81 positioning at the various quasi-equivalent locations more clearly (see subsection “Decoration proteins employ a variety of binding schemes to adhere to phage capsids”).

One might have presumed that the most angled positions that represent a "weak point in many capsids" would be the one where Dec binds. There does not seem to be much discussion of this interesting point, which may point to Dec's function – perhaps not to stabilize capsids after all?

We have revised this statement to more clearly explain that Dec does bind to the capsids where they suffer from AFM damage and are thus a weak point:

“Biophysical measurements have shown that Dec plays a much larger role in stabilizing dsDNAcontaining heads compared to empty capsids, and atomic force microscopy experiments showed Dec indeed binds capsids at positions most prone to mechanical damage (i.e. quasi three-fold sites between hexons) [Llauro et al., 2016]”

5) Also, in Figure 6, the arc with the denotation 54.2Å is confusing. What is this the distance between? It seems to indicate an angle rather than a distance. Was it meant to be 54.2°? It might be helpful to list these angles differences between all three types of trimers.

The intention of the distance measurement was to emphasize the large difference in curvature. However, we appreciate that this was confusing and have since removed this label in the new figure. We hope that the new figure is clearer and that the presence of the Dec density sufficiently emphasizes the distance without the need for listing additional direct measurements. We have also updated the figure legend accordingly.

6) I find it a really missed opportunity to demonstrate the claimed quality of the maps by not showing some region with excellent side-chain density with the fold model fit into it – not in the manuscript, supplementary figures or videos.

We added an enlarged area view to Figure 1 showing the fit of side chains into the cryo-EM density map of the capsid protein spine helix.

7) This rebuttal statement "We have de-emphasized the notion of discriminatory binding in our title and throughout the text" does not seem to be accurate. The Abstract states: "The Dec protein from the P22-like phage L has a highly unusual binding strategy that precisely distinguishes between nearly identical three-fold and quasi-three-fold sites of the icosahedral capsid." and "Here, we investigate how Dec is able to distinguish subtly different capsid binding sites with high specificity." This aspect has rather more prominence than claimed.

We have deleted the word “precisely” in the Abstract. We also deleted the statement “Here, we investigate how Dec is able to distinguish subtly different capsid binding sites with high specificity.”

Additionally, we have revised the next sentence to read: “We report the structure of phage L Dec protein, which has a novel fold for a decoration protein and propose an explanation for how Dec may be able to bind to subtly different capsid binding sites.”